# Structural basis of malodour precursor transport in the human axilla

**Gurdeep S Minhas[1†], Daniel Bawdon[2†], Reyme Herman[2], Michelle Rudden[2], Andrew P Stone[2], A Gordon James[3], Gavin H Thomas[2]\*, Simon Newstead[1]\***

[1]Department of Biochemistry, University of Oxford, Oxford, United Kingdom; [2]Department of Biology, University of York, York, United Kingdom; [3]Unilever Discover, Bedford, United Kingdom

**Abstract** Mammals produce volatile odours that convey different types of societal information. In *Homo sapiens*, this is now recognised as body odour, a key chemical component of which is the sulphurous thioalcohol, 3-methyl-3-sulfanylhexan-1-ol (3M3SH). Volatile 3M3SH is produced in the underarm as a result of specific microbial activity, which act on the odourless dipeptide-containing malodour precursor molecule, S-Cys-Gly-3M3SH, secreted in the axilla (underarm) during colonisation. The mechanism by which these bacteria recognise S-Cys-Gly-3M3SH and produce body odour is still poorly understood. Here we report the structural and biochemical basis of bacterial transport of S-Cys-Gly-3M3SH by *Staphylococcus hominis*, which is converted to the sulphurous thioalcohol component 3M3SH in the bacterial cytoplasm, before being released into the environment. Knowledge of the molecular basis of precursor transport, essential for body odour formation, provides a novel opportunity to design specific inhibitors of malodour production in humans.

DOI: https://doi.org/10.7554/eLife.34995.001

**\*For correspondence:**
gavin.thomas@york.ac.uk (GHT);
simon.newstead@bioch.ox.ac.uk (SN)

[†]These authors contributed equally to this work

## Introduction

In *Homo sapiens* the skin of the underarm (axilla) provides a unique niche for bacteria. Through the secretions of various glands, that open onto the skin or into hair follicles, this environment is nutrient-rich and hosts a unique microbial community (*Taylor et al., 2003*). Esterified fatty acids and other lipids are secreted by the sebaceous gland, while the eccrine glands produce a dilute salt solution with lactate and other organic solutes being secreted (*Bovell, 2015*). A third exocrine gland, the apocrine, which is more specialized and found only in the axilla, areolae, genitalia and external auditory meatus (ear canal) (*Collins, 1989*), starts secreting an odourless lipid-rich viscous secretion with the onset of puberty (*Figure 1A*). The lipid rich secretion is not involved in thermoregulation like the eccrine gland and in mammals has a likely role in scent generation (*Collins, 1989*).

In humans, the link between the axillary apocrine gland, underarm bacteria and body odour (BO), was first recognised over 50 years ago (*Shelley et al., 1953*) and early culture-based studies on this relationship found the axillary microbiota to be dominated by Staphylococcus, Corynebacterium and Propionibacterium species, with the *Corynebacteria* in particular correlating with malodour intensity (*Taylor et al., 2003*; *Leyden et al., 1981*). More recent culture-independent meta-taxonomic studies have confirmed that the axilla is dominated by these genera, although the presence of additional taxa not previously indicated by culture-based methods is also evident, notably Gram-positive anaerobic cocci (GPAC) belonging to the *Anaerococcus* and *Peptoniphilus* genera (*Costello et al., 2009*; *Egert et al., 2011*; *Troccaz et al., 2015*). Various molecules produced in the axilla have been linked to malodour, including 16-androstene steroids (*Gower et al., 1994*) and short-chain (C2-C5) volatile fatty acids (VFAs) produced by axillary bacteria via several metabolic routes (*James et al., 2004*). However, 16-androstenes are now believed to be only minor contributors to axillary malodour

**eLife digest** Human body odour contains a number of chemicals, but the most pungent and recognisable are thioalcohols. These molecules are created through a series of chemical reactions that start with an odourless precursor, a compound produced in glands located in our armpits. Then, a type of bacteria called *Staphylococcus hominis* takes in these molecules and transforms them into smelly thioalcohols. The precise details of how the bacteria do this are not clear.

Now, Minhas, Bawdon et al. show how *S. hominis* uses a transport protein in its membrane to bring the odourless precursor inside. In the experiments, tools such as X-ray crystallography captured snapshots of this transporter as it was moving the compound into the bacteria. This helped to understand how the bacteria recognize these precursors, as well as the exact structure of these molecules and of their transporters.

The experiments also reveal that bacteria which do not create odour can also ingest the precursors through this same process. This suggests that the odour production is a unique process that happens once these molecules are inside *S. hominis.*

The findings imply that humans and their body odour-producing bacteria likely evolved together. In other mammals, the bacterial production of bodily smells is linked to the release of pheromones, which are chemicals involved in communication and in selecting sexual partners. It is not clear whether this is also true for humans. Ultimately, learning more about how *S. hominis* converts precursor molecules into thioalcohols could lead to new ways of nipping body odours in the bud.

DOI: https://doi.org/10.7554/eLife.34995.002

(*James et al., 2013*), and it is accepted that the VFAs most heavily involved in BO are a series of structurally-unusual medium-chain (C6-C10) acids originating from *N*-acylglutamine precursors present in apocrine sweat (*Natsch et al., 2003*; *Natsch et al., 2006*). A third class of molecule, the thioalcohols (also called sulfanylalkanols), have recently been identified as important components of human body odour and a number of these have been detected within the volatiles released from the underarm (*Troccaz et al., 2004*; *Natsch et al., 2004*; *Hasegawa et al., 2004*). They are particularly pungent and can be detected in $pg.l^{-1}$ quantities, an order of magnitude lower than many other volatile chemicals emanating from the skin (*Natsch et al., 2004*). The most abundant and pungent of these thioalcohols is 3-methyl-3-sulfanylhexan-1-ol (3M3SH) (*Figure 1B*), which is a primary causal molecule of axillary malodour and subsequent BO (*Troccaz et al., 2004*; *Hasegawa et al., 2004*).

Thioalcohols are known to be released as a peptide-conjugated precursor from the apocrine gland, which secretes directly into the hair follicle (*Figure 1A*). The biosynthetic origin of the thioalcohol precursors is currently poorly understood, but they are thought to be secreted into the axilla via the ABC transporter ABCC11 as odourless glutathione conjugates (*Baumann et al., 2014a*). The glutathione conjugated form (SG-3M3SH) has never been detected in axillary sweat; however, the L-cysteinylglycine dipeptide conjugated form, S-[1-(2-hydroxyethyl)−1-methylbutyl]-L-cysteinylglycine (S-Cys-Gly-3M3SH) (*Figure 1B*), is readily detectable (*Starkenmann et al., 2005*) and recently a γ-glutamyl transferase (GGT1) has been demonstrated to localise to the apocrine sweat gland and convert SG-3M3SH to S-Cys-Gly-3M3SH (*Baumann et al., 2014a*) (*Figure 1C*). We and others have demonstrated that S-Cys-Gly-3M3SH can be taken up and metabolised by a highly limited range of staphylococcal species present in the axilla, namely *Staphylococcus hominis*, *Staphylococcus haemolyticus* and *Staphylococcus lugdunensis*, to liberate 3M3SH, that subsequently exits the cell through an undefined mechanism (*Troccaz et al., 2004*; *Starkenmann et al., 2005*; *Bawdon et al., 2015*). A strong correlation between *S. hominis* and body odour production has recently been established using culture-independent taxonomic studies of the axillary microbiota (*Troccaz et al., 2015*), confirming the biological significance of this low-abundance species within the axilla in BO production in humans. However, the mechanism by which the S-Cys-Gly-3M3SH precursor molecule is taken up into *S. hominis* and related thioalcohol producers is currently unknown, hampering efforts to identify novel mechanisms to disrupt this process and control odour production in humans.

Here we identify the biochemical and structural basis for the recognition and specific transport of the malodour precursor peptide, S-Cys-Gly-3M3SH, into *S. hominis*. We first show that S-Cys-Gly-3M3SH is actively taken up by *S. hominis* through a specific secondary active transporter,

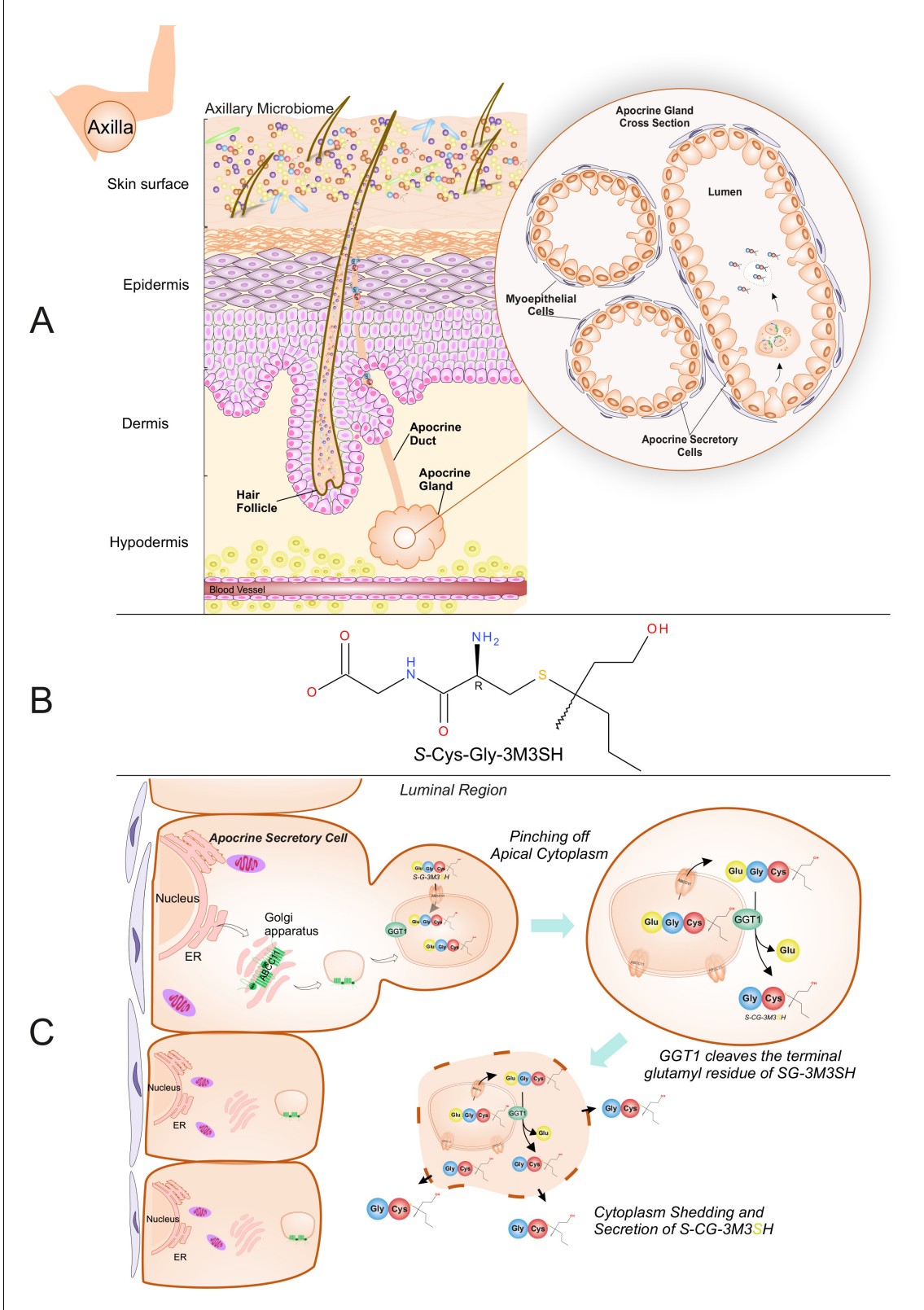

**Figure 1.** Biochemical route of axillary malodour precursor peptide S-Cys-Gly-3M3SH. (**A**) Schematic representation of the axilla. The surface of the skin is composed of the axillary microbiome, typically dominated by either *Corynebacterium* or Staphylococci. Axillary apocrine glands are subcutaneously located in the hypodermis. Apocrine glands secrete a variety of odourless precursors that are bio-transformed into odorous volatiles by the axillary microbiome. Sulfanylalkanols (e.g. 3-methyl-3-sulfanylhexan-1-ol (3M3SH)) are secreted as cysteinylglycine conjugated peptides (***Starkenmann et al.,***

*Figure 1 continued*

*2005*; *Emter and Natsch, 2008*). *Staphylococcus hominis* transport and metabolise cysteinylglycine- 3-methyl-3-sulfanylhexan-1-ol (*S*-Cys-Gly-3M3SH) to liberate the odorous volatile 3M3SH, which is one of the major components of axillary malodour. (**B**) Chemical structure of the malodour precursor peptide, *S*-Cys-Gly-3M3SH. (**C**) Synthesis of *S*-Cys-Gly-3M3SH. *S*-Cys-Gly-3M3SH is derived from the glutathione 3-methyl-3-sulfanylhexanol conjugate (SG-3M3SH). SG-3M3SH is actively transported by the ATP-binding cassette transporter sub-family C member 11 (ABCC11) into apocrine secretory vesicles (*Baumann et al., 2014b*). In the secretory vesicle, *S*-Gly-3M3SH is bio-transformed by γ-glutamyl transferase 1 (GGT1), which cleaves the terminal glutamyl residue generating S-Cys-Gly-3M3SH. S-Cys-Gly-3M3SH is secreted to the surface of the skin by apocrine secretion, which characteristically involves pinching off the apical cytoplasm and partial shedding releasing intracellular components into the luminal region of the apocrine gland. Upon release the cytoplasm may be recovered and repaired and a new cycle of secretion can occur (*Farkaš, 2015*).

DOI: https://doi.org/10.7554/eLife.34995.003

STAH0001_1446 (SH1446), a member of the proton coupled oligopeptide transporter (POT) family (*Hagting et al., 1994*; *Paulsen and Skurray, 1994*). Through a combination of structural, biochemical and cell based assays we reveal how this peptide transporter is able to recognise a thioalcohol-conjugated peptide and further, show that transport is coupled to the inwardly directed proton electrochemical gradient. Our results suggest that SH1446 is a clear target for inhibitor design to prevent malodour production, thus providing further molecular insights into the role of the skin microbiota in human scent generation.

## Results

### POT transporters are essential for transport of thiol-conjugated dipeptides in *E. coli*

To elucidate how thiol-conjugated dipeptides are transported by bacteria, we first examined the model organism *Escherichia coli*, which is able to recognise and metabolise the dipeptide version of the non-physiological malodour precursor *S*-benzyl-L-cysteine, namely, S-benzyl-L-cysteinylglycine (*James et al., 2013*). Resting cells of *E. coli* were indeed able to remove S-benzyl-L-cysteinylglycine from solution and break it down to release the thiol active product, benzyl mercaptan (*Figure 2— figure supplement 1A,B*). Following transport into the cell, the dipeptide must be cleaved to release the benzyl mercaptan, most likely through a cysteine-*S*-conjugate β-lyase-type activity, for which a number have been described in *E. coli*, including MetC and MalY (*Dwivedi et al., 1982*; *Zdych et al., 1995*; *Awano et al., 2003*). Interestingly, TnaA which is a tryptophanase known to have L-cysteine desulfhydrase activity (*Awano et al., 2005*) is also capable of releasing benzyl mercaptan.

Individual disruption of *metC* and *malY* did not decrease the amount of benzyl mercaptan produced by *E. coli*, while disruption of *tnaA* led to an almost complete loss of biotransformation (*Figure 2—figure supplement 1C*). This suggests that *E. coli* takes up the S-benzyl-L-cysteinylglycine, whereupon a host dipeptidase cleaves the glycine residue prior to cleavage of the *S*-benzyl-cysteine by TnaA to release the benzyl mercaptan.

To discover the genetic basis of the transport and metabolism of these thiol-conjugated dipeptides, we exploited the knock out (KO) library of single gene disruptions in *E. coli* K-12 (*Baba et al., 2006*). We hypothesised that the dipeptide moiety might direct the molecule through a peptide transporter. In *E. coli* the main peptide systems are relatively well characterised, the two main transport systems belong to the ATP binding cassette (ABC) family and the proton dependent oligopeptide transporter (POT) family (*Figure 2B*). The *E. coli* Opp and Dpp ABC transporters have been extensively studied in *E. coli* and the closely related bacterium *Salmonella typhimurium* (*Smith et al., 1999*; *Klepsch et al., 2011*; *Abouhamad et al., 1991*; *Goodell and Higgins, 1987*; *Tame et al., 1995*) and an *E. coli* BW25113 strain with non-functional Opp and Dpp system (ΔDB1), was used to assess the role of these transporters in the production of benzyl mercaptan. Surprisingly, benzyl mercaptan production was similar to the wild-type strain, suggesting that neither ABC transporter was important for Cys-Gly-conjugate transport (*Figure 2A*). Having ruled out transport via the ABC peptide transporters, we then broadened our search to include additional peptide transporters that have been described in *E. coli*. These were the four members of the proton dependent oligopeptide transporter, or POT family, encoded by *dtpA-D*, that have poorly defined physiological roles during

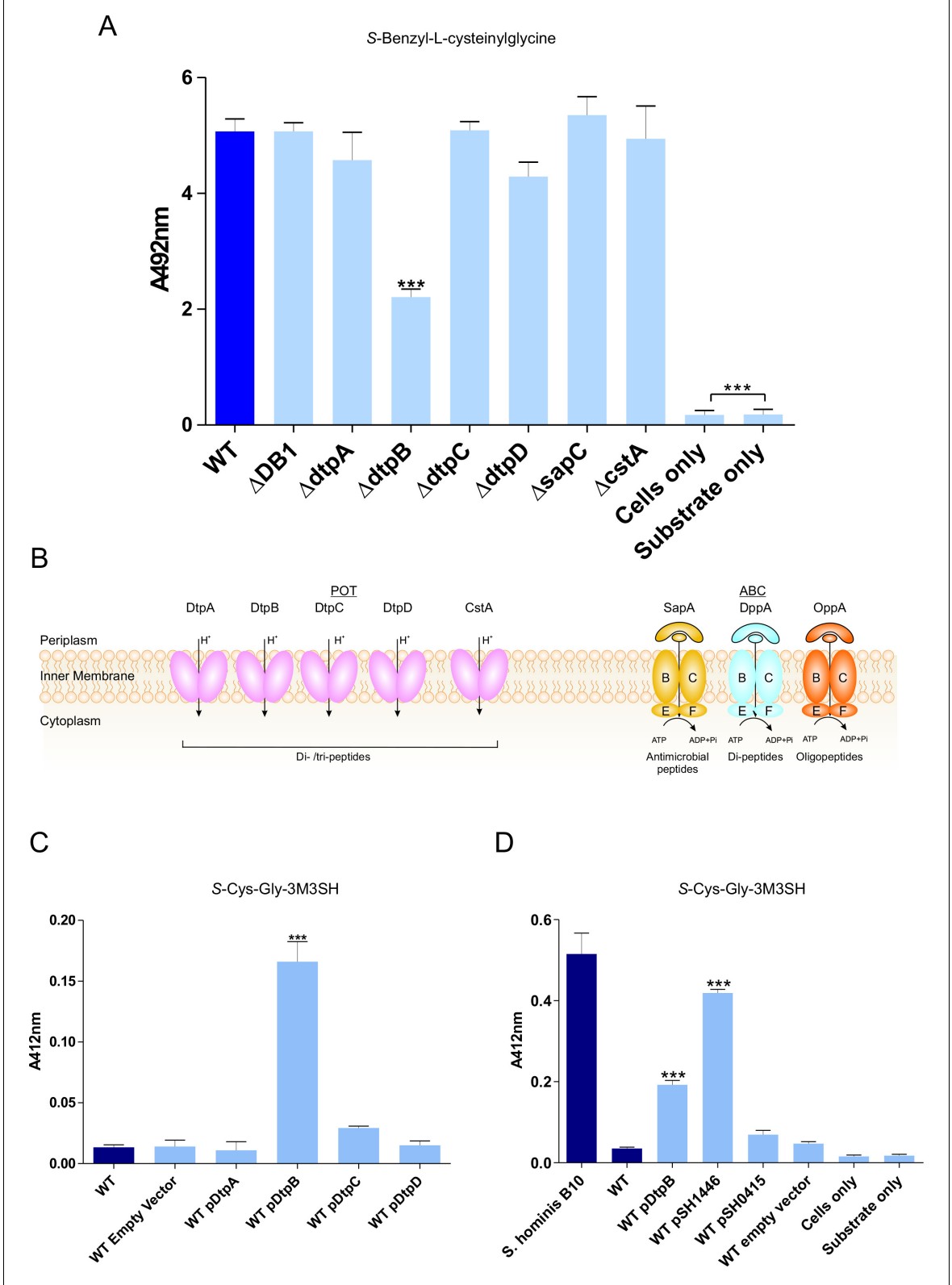

**Figure 2.** Biotransformation of both model (S-Benzyl-L-cysteinylglycine) and physiological substrate *S*-Cys-Gly-3M3SH by model organism *E.coli* K-12. (A) Biotransformation of S-Benzyl-L-cysteinylglycine by *E. coli* strains containing deletion in known or predicted peptide transporters. Strain ΔDB1 is an *oppA dppA* double mutant. Thioalcohol yield ($A_{492}$) was quantified for each strain using MTS/PMS following 4 hr incubation. (B) Schematic overview of the main peptide transport systems in *E. coli*. (C) Biotransformation of *S*-Cys-Gly-3M3SH in *E. coli* K-12 overexpressing Dtp POT transporter proteins.

*Figure 2 continued*

(D) S-Cys-Gly-3M3SH biotransformation by *E. coli* K-12 overexpressing *S. hominis* STAHO0001_1446 (pSH1446) and STAHO0001_0415 (pSH0415). *E. coli* overexpressing *dtpB* (pDtpB) is shown for reference. Thioalcohol yield ($A_{412}$) was quantified using DTNB after 24 hr incubation. Cultures with overexpressing plasmids were pre-induced with 0.0001% L-arabinose overnight. E. coli expressing pBADcLIC2005 with no gene insertion was included as a negative control (Empty vector). Error bars: ±SD (biological triplicates); *p<0.05, **p<0.01, ***p<0.001; One-way ANOVA followed by Dunnett's Multiple Comparison test.

DOI: https://doi.org/10.7554/eLife.34995.004

The following figure supplements are available for figure 2:

**Figure supplement 1.** Uptake and biotransformation of thiol-conjugated dipeptides by *E.coli* K-12.

DOI: https://doi.org/10.7554/eLife.34995.005

**Figure supplement 2.** DtpB increases biotransformation of S-Benzyl-L-cysteinylglycine in *E. coli* K-12, while *S. hominis* transporter SH1446 does not.

DOI: https://doi.org/10.7554/eLife.34995.006

**Figure supplement 3.** Phylogenetic relationship of staphylococcal SH1446 orthologues.

DOI: https://doi.org/10.7554/eLife.34995.007

**Figure supplement 4.** Inhibition of CG-3M3SH biotransformation in the presence of L-Ala peptides.

DOI: https://doi.org/10.7554/eLife.34995.008

**Figure supplement 5.** L-Ala peptides do not inhibit in vitro cleavage of Cys-3M3SH by TnaA.

DOI: https://doi.org/10.7554/eLife.34995.009

growth on peptides (*Harder et al., 2008*; *Prabhala et al., 2014*; *Casagrande et al., 2009*), the CstA protein, orthologues of which have been demonstrated to be peptide transporters (*Garai et al., 2016*) and the Sap system, which is implicated in uptake of cationic antimicrobial peptides (*Groisman et al., 1992*). Most of these strains had similar levels of benzyl mercaptan production to the wild-type, with the exception of *dtpB* which had an overall 50% drop in thiol production (*Figure 2A*). These data suggest that transporters belonging to the POT family are most likely responsible for Cys-Gly-conjugate transport, revealing a new biological role for bacterial POT transporters in the uptake of Cys-Gly-(S) conjugates.

## Identification of a POT transporter from *S. hominis* that confers malodour production to *E. coli*

Having established that *E. coli* has the ability to transport and cleave a Cys-Gly-(S) conjugated substrate, we reasoned that overexpression of a POT transporter might confer the ability to recognise the physiological malodour precursor S-Cys-Gly-3M3SH. The four *dtp* genes were cloned into a pBAD vector commonly used for overexpression of membrane transporters (*Mulligan et al., 2009*; *Mulligan et al., 2012*) and transformed into the wild-type *E. coli* background. The ability of the resulting strains to convert S-Cys-Gly-3M3SH to 3M3SH was subsequently measured (*Bawdon et al., 2015*). Consistent with the reduction in benzyl mercaptan production reported in *Figure 2A*, the *dtpB* gene alone was able to confer a malodour production (BO$^+$) phenotype to *E. coli*, which was not seen with any of the other *dtp* genes (*Figure 2C*). The same pattern was observed with S-benzyl-Cys-Gly as a substrate (*Figure 2—figure supplement 2B*), suggesting that DtpB has a unique ability to transport these unusual conjugated peptides.

The molecular basis for S-Cys-Gly-3M3H transport into malodour producing staphylococci is unknown, and the genetic intractability of these poorly studied coagulase negative *Staphylococcal* species makes it difficult to study using genetic approaches. We therefore used an alternative approach exploiting our discovery that expression of a transporter in trans could produce a BO$^+$ phenotype to *E. coli*. Given our discovery that *E. coli* POT family peptide transporters can transport S-Cys-Gly-3M3SH, we examined the genome of *S. hominis* SK1119, as *S. hominis* are the most abundant malodour producing Staphylococcal species in the underarm microbiota. Unlike *E. coli*, *S. hominis* contains only a single POT family member DtpT (SH1446), orthologues of which were seen in all sequenced staphylococci (*Figure 2—figure supplement 3A*). The POT peptide systems remain largely uncharacterised in staphyloloccci, although in the well-studied pathogen *S. aureus*, the single POT transporter encoded by *dtpT* is the sole route for dipeptide uptake in this species (*Hiron et al., 2007*). We also identified a second system, encoded by STAHO0001_0415 (SH0415), which is annotated as an MFS transporter and only present in *S. hominis* and *S. haemolyticus*, two known thioalcohol producers, and which is genetically linked to genes involved in peptide and amino acid

metabolism (*Figure 2—figure supplement 3B*). Both transporters were cloned into pBADcLIC2005 and transformed into *E. coli* for functional analysis. When levels of 3M3SH production were measured in these strains, uptake was only enhanced by the presence of SH1446, with SH0415 having little effect on 3M3SH activity (*Figure 2D*). In contrast, with S-benzyl-cysteinlyglycine as a substrate, there was no increase in production of benzyl mercaptan with SH1446 but a significant increase was observed with SH0415 (*Figure 2—figure supplement 2B*). Together these data strongly support the conclusion that the S-Cys-Gly-3M3SH transporter in *S. hominis* is the POT transporter STAH0001_1446 (SH1446).

To further demonstrate the importance of SH1446 in the physiological transport of *S*-Cys-Gly-3M3SH we used a biochemical approach to examine whether peptides could inhibit 3M3SH production by competing with *S*-Cys-Gly-3M3SH uptake. Firstly, we used our *E. coli* system where 3M3SH production is seen when SH1446 is expressed in trans and we observed clear inhibition of 3M3SH production with both di- and tri-Ala peptides (*Figure 2—figure supplement 4*) but not with L-Ala alone nor tetra-Ala. The purified TnaA enzyme that cleaves Cys-3M3SH was not inhibited by any of these peptides (*Figure 2—figure supplement 5*) , supporting the hypothesis that the whole cell inhibition is due to inhibition of peptide transport. Similarly, we measured the ability of the same peptides to inhibit 3M3SH production directly in *S. hominis*  (*Figure 2—figure supplement 4* ) and observed a comparable profile. It should be noted that a significant affect was observed for 25 mM tetra-Ala in *S. hominis* wild-type, however this affect is due to the acidic pH that was required to dissolve the peptide which appears to inhibit metabolism in *S. hominis* but not *E. coli*.

## The *S. hominis* POT transporter, PepT$_{Sh}$ transports *S*-Cys-Gly-3M3SH

POT transporters belong to the Major Facilitator Superfamily (MFS) and use an inwardly directed proton electrochemical gradient to drive the concentrative uptake of di- and tri-peptides across the cell membrane for metabolic assimilation (*Daniel and Kottra, 2004*). To establish SH1446 as the protein responsible for S-Cys-Gly-3M3SH recognition and transport, the gene was cloned and overexpressed in *E. coli* for functional and structural studies. The purified protein, hereafter referred to as PepT$_{Sh}$ (Peptide Transporter *Staphylococcus hominis*) was purified and reconstituted into liposomes. Transport was studied using a previously employed pyranine based reporter assay, which measures transport via changes to the internal pH of the liposomes (*Parker et al., 2014*) (*Figure 3A*). Acidification of the lumen of the liposome was observed in the presence of a membrane potential ($\Delta\Psi$, negative inside), and addition of external di- or tri-peptide substrate, demonstrating that PepT$_{Sh}$ is a proton coupled peptide transporter. Importantly we also observed acidification in the presence of the S-Cys-Gly-3M3SH precursor peptide, confirming our earlier identification of PepT$_{Sh}$ as the transporter for this molecule in *S. hominis*.

To understand the recognition of S-Cys-Gly-3M3SH compared to natural peptides, we determined IC$_{50}$ values using competition with $^{3}$H-di-alanine. The IC$_{50}$ value for tri-alanine, di-alanine and Cys-Gly were 23.7 $\pm$ 3.4, 72.2 $\pm$ 9.7 and 115 $\pm$ 22.2 µM, respectively (*Figure 3B*), indicating that PepT$_{Sh}$ is able to recognise tri-Ala more favourably than di-Ala. However, for S-Cys-Gly-3M3SH the IC$_{50}$ value was 362 $\pm$ 21.8 µM, three-fold higher than the Cys-Gly peptide. Our data therefore show that while PepT$_{Sh}$ can transport the modified peptide, the addition of the thioalcohol group on the cysteine side chain has substantially impaired the recognition and transport versus natural peptides.

## Crystal structure of PepT$_{Sh}$ bound to *S*-Cys-Gly-3M3SH

To gain further insight into the mechanism of substrate binding we successfully obtained a high-resolution co-crystal structure of PepT$_{Sh}$ at 2.5 Å resolution, bound to S-Cys-Gly-3M3SH using the *in meso* crystallisation method (*Figure 4A*, *Table 1* and *Figure 4—figure supplement 1*). PepT$_{Sh}$ adopts the canonical MFS fold with helices H1-H6 forming the N-terminal bundle and H7-H12 the C-terminal bundle. PepT$_{Sh}$ was crystallised in an inward open state, observed previously for other members of the POT family (*Newstead, 2015*), wherein the extracellular gate, consisting of helices H1-2 and H7-8 from the N- and C-terminal bundles respectively, is closed and the intracellular gate, consisting of helices H4-5 and H10-11 is open. Similar to other bacterial POT family members, PepT$_{Sh}$ contains two additional helices inserted between the N- and C-terminal bundles, termed HA and HB (*Figure 4B* and *Figure 4—figure supplement 2*). Unlike previous POT structures however, PepT$_{Sh}$ also contains a well-structured β-hairpin motif between helices H7 and H8 on the

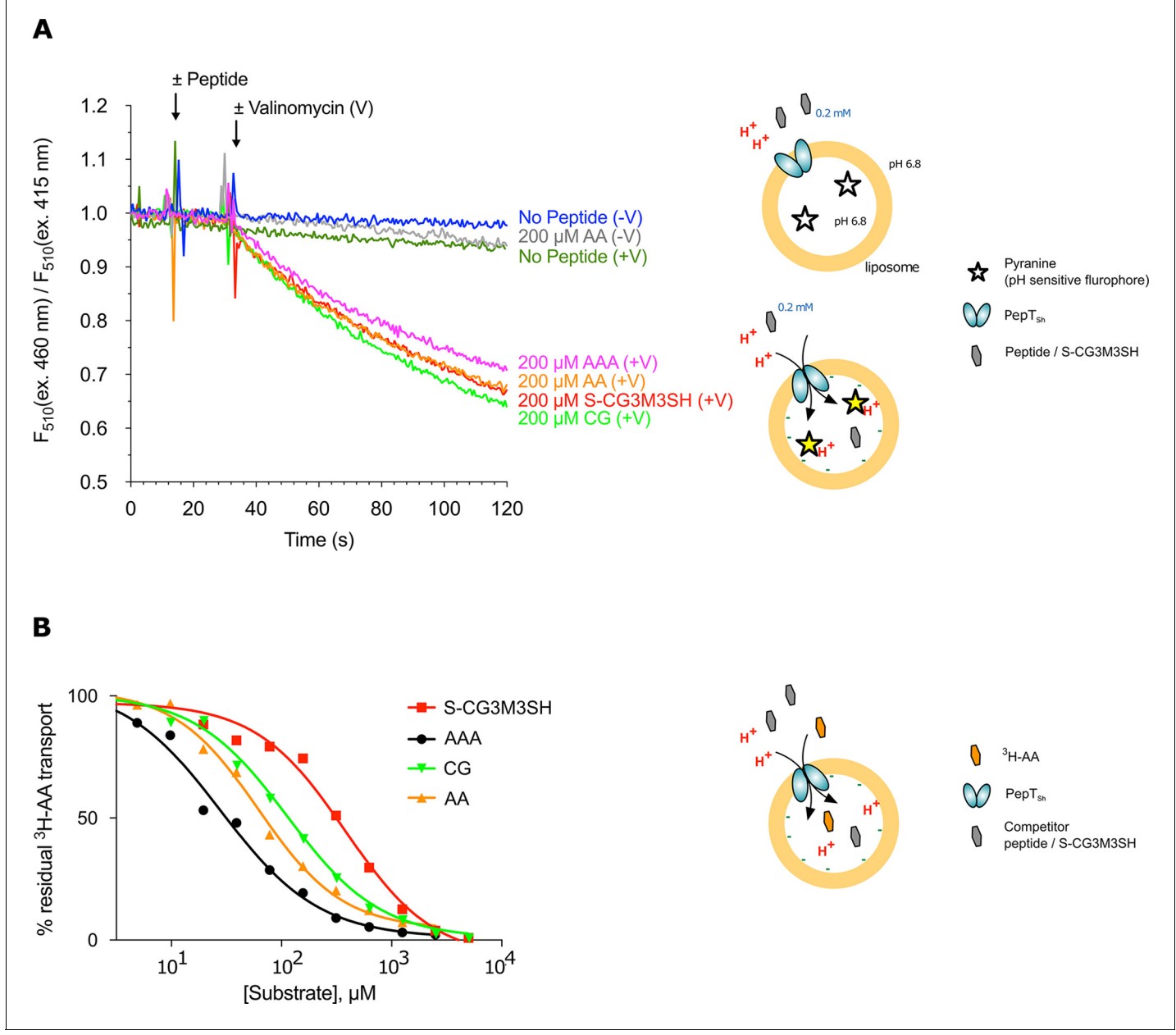

**Figure 3.** Functional characterisation of PepT$_{Sh}$. (A) Monitoring proton coupled peptide uptake using a pH sensitive dye. PepT$_{Sh}$ was reconstituted into liposomes loaded with pyranine and a high concentration of potassium ions. The external solution contains either peptide, Ala-Ala (orange), Ala-Ala-Ala (purple), Cys-Gly (Green) or S-Cys-Gly-3M3SH (red) and a low potassium concentration. On addition of valinomycin (V) the membrane becomes highly permeable to potassium, generating a hyperpolarised membrane potential (negative inside), which drives the uptake of protons and ligand, protonating the pyranine dye and altering its fluorescent properties. A change in the ratio of fluorescence demonstrates active transport of the ligand. (B) Representative IC$_{50}$ curves showing the relative affinity of the different ligands shown in A for PepT$_{Sh}$. Assays were set up to measure the ability of competitor peptides to reduce uptake of tritiated di-Alanine (AA) peptide.

DOI: https://doi.org/10.7554/eLife.34995.010

extracellular side of the membrane, extending out in line with the lipid head groups. Interestingly we observed a number of monoolein lipid molecules that cluster around both the β- hairpin and the cavity created between helices HA and HB with the transporter domain. The role of helices HA and HB in prokaryotic POT family transporters still remains unclear. However, the observation of lipid molecules bound between these helices and the transport domain suggests a possible role in

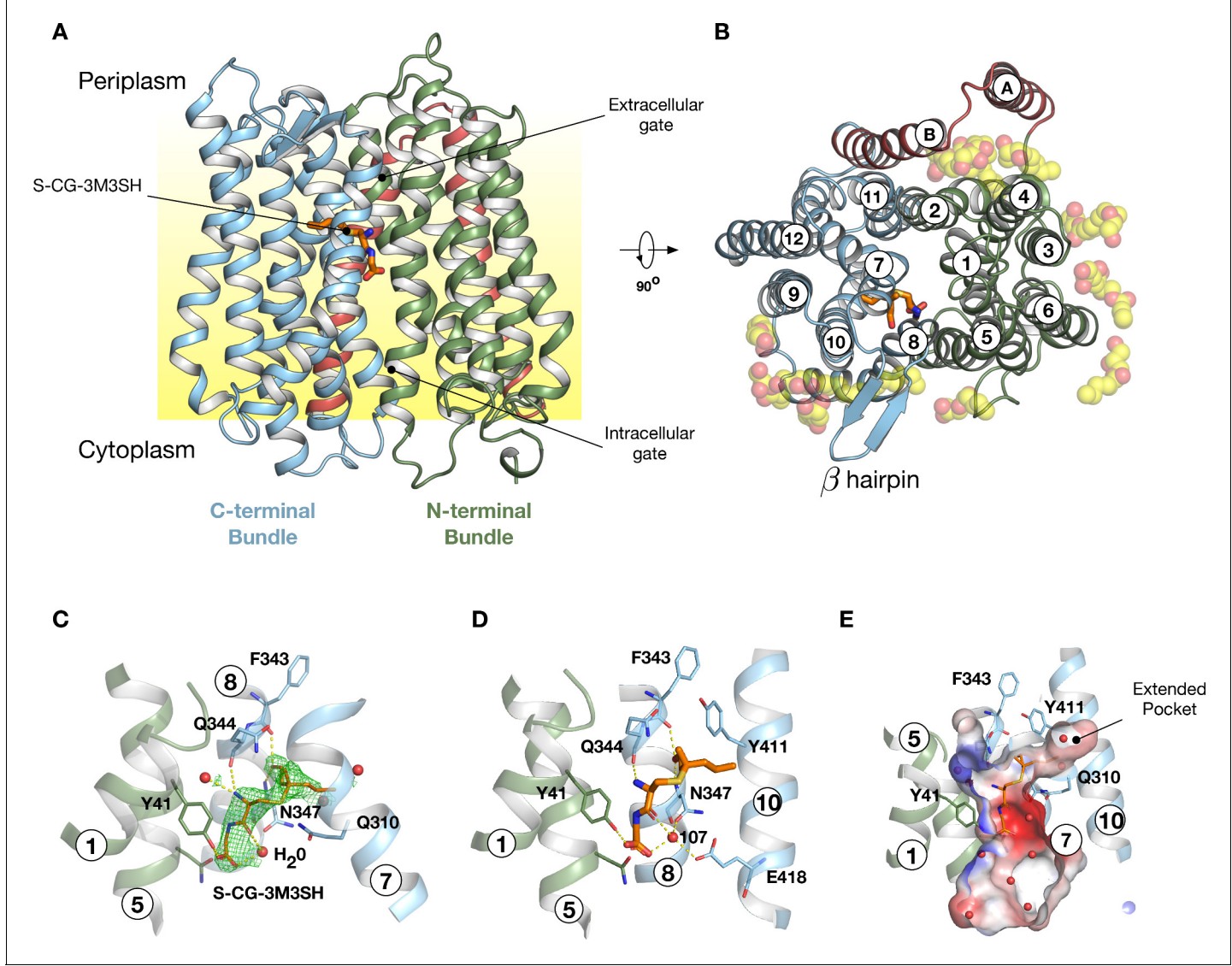

**Figure 4.** Crystal structure of PepT$_{Sh}$. (**A**) Crystal structure of PepT$_{Sh}$ shown in the plane of the membrane. Transmembrane helices making up the N-terminal bundle, H1-H6, are coloured green and pack against the C-terminal bundle, helices H7-H12. The bound S-Cys-Gly-3M3SH peptide is shown in sticks. (**B**) View of A rotated 90 °. The additional helices, HA and HB, can be clearly observed packing to one side of the transporter. The bound lipids are shown as spheres. (**C**) Zoomed in view of the binding site showing the bound S-Cys-Gly-3M3SH ligand with the mFo-DFc difference electron density map contoured at 3 σ in green mesh. (**D**) Equivalent view of the binding site showing the location of ordered water molecules in the binding site. (**E**) View showing the electrostatic surface of the binding site of the transporter with the hydrophobic pocket accommodating the acyl chain of the ligand.

DOI: https://doi.org/10.7554/eLife.34995.011

The following figure supplements are available for figure 4:

**Figure supplement 1.** Final refined electron density map for PepT$_{Sh}$.
DOI: https://doi.org/10.7554/eLife.34995.012

**Figure supplement 2.** Sequence alignment of PepT$_{Sh}$ with bacterial species found in the human axilla and human peptide transporters.
DOI: https://doi.org/10.7554/eLife.34995.013

**Figure supplement 3.** Structure of PepT$_{Sh}$ bound to the modified peptide S-Cys-Gly-3M3SH.
DOI: https://doi.org/10.7554/eLife.34995.014

**Figure supplement 4.** Comparison between the binding position of S-Cys-Gly-3M3SH and the natural di- and tri-peptides co-crystallised with PepT$_{St}$.
DOI: https://doi.org/10.7554/eLife.34995.015

**Table 1.** Data collection and refinement statistics.
Statistics for the highest-resolution shell are shown in parentheses.

| | PepT$_{sh}$ (PDB: 6EXS) |
|---|---|
| Wavelength | 0.8729 |
| Resolution range | 45.1–2.5 (2.589–2.5) |
| Space group | P 1 21 1 |
| Unit cell | 60.04 55.576 100.866 90 104.749 90 |
| Total reflections | 151797 (14474) |
| Unique reflections | 22501 (2204) |
| Multiplicity | 6.7 (6.6) |
| Completeness (%) | 99.43 (99.73) |
| Mean I/sigma (I) | 9.76 (1.48) |
| Wilson B-factor | 43.87 |
| R-merge | 0.159 (1.28) |
| R-meas | 0.172 (1.39) |
| R-pim | 0.07 (0.54) |
| CC1/2 | 0.997 (0.59) |
| CC* | 0.999 (0.86) |
| Reflections used in refinement | 22504 (2204) |
| Reflections used for R-free | 1137 (110) |
| R-work | 0.21 (0.21) |
| R-free | 0.23 (0.24) |
| CC (work) | 0.88 (0.69) |
| CC (free) | 0.85 (0.52) |
| Number of non-hydrogen atoms | 4063 |
| macromolecules | 3614 |
| ligands | 384 |
| solvent | 65 |
| Protein residues | 487 |
| RMS(bonds) | 0.01 |
| RMS(angles) | 1.04 |
| Ramachandran favored (%) | 98.35 |
| Ramachandran allowed (%) | 1.65 |
| Ramachandran outliers (%) | 0 |
| Rotamer outliers (%) | 1.47 |
| Clashscore | 2.59 |
| Average B-factor | 57.0 |
| macromolecules | 54.72 |
| ligands | 80.53 |
| solvent | 51.2 |

DOI: https://doi.org/10.7554/eLife.34995.016

stabilising these transporters in the membrane, which may be an adaption specific to prokaryotic members of the POT family (*Newstead et al., 2011*).

Following extensive screening and co-crystallisation we observed clear m$F_o$-D$F_c$ difference electron density for the S-Cys-Gly-3M3SH ligand in the peptide binding site (*Figure 4C* and *Figure 4—*

*figure supplement 3*). The Cys-Gly part of the precursor is positioned such that the carboxy termini of the peptide faces towards the intracellular side of the binding site. The terminal carboxyl group makes two hydrogen bond interactions, one with Tyr41 on H1 and a second interaction to a water molecule (W107), which links the carboxyl group with Asn347 on H8. The carbonyl group of the CG peptide also makes a hydrogen bond to this water molecule, creating a well-coordinated interaction network between the peptide and Asn347 (*Figure 4D*). The terminal amino group makes two further polar interactions, with a water molecule that sits next to Tyr41 and also to the carbonyl group of Gln344, which sits further up on H8. In contrast, the thioalcohol moiety points towards the extracellular gate, with the sulphur atom sitting close to Gln310 on H7, whereas the alcohol group extends towards helix H8 making interactions with backbone carbonyl and amide groups of Phe343 and Asn347, respectively. The acyl group extends into a large pocket nestled between helices H7 and H10 (*Figure 4E*). This long pocket (~10 Å) extends from the central peptide binding site into the C-terminal bundle of the transporter and is both hydrophobic and polar in character.

Previously, we showed that a related POT family transporter, PepT$_{St}$, is able to bind peptides in different orientations in the binding site, with a di-peptide adopting a horizontal position and a tri-peptide a vertical one (*Lyons et al., 2014*) and suggested this flexibility may be responsible for the accomodation of larger peptide ligands (*Newstead, 2017*). Here we see that the *S*-Cys-Gly-3M3SH peptide adopts a more vertical orientation, similar to the tri-alanine position in PepT$_{St}$. However, there are significant differences in the interactions. A structural overlay of the *S*-Cys-Gly-3M3SH peptide shows that it sits much further up in the binding site compared to the natural peptides captured in PepT$_{St}$ (*Figure 4—figure supplement 4A & B*), most likely as a result of the requirement to accommodate the thioalcohol group in the extended pocket. This has a knock-on effect with respect to the position of the amino and carboxy termini. In PepT$_{St}$, the amino terminal group of the di-peptide directly interacts with helix H8 and H10, whereas the tri-peptide makes a much weaker interaction through a carbonyl group. In contrast, in PepT$_{Sh}$ this interaction is achieved through a water mediated hydrogen bond (*Figure 4D*).

The role of water in mediating interactions between peptides and their binding sites has been previously observed in the ABC transporter binding domains (*Tame et al., 1995*), albeit for side chain independent binding. A similar mechanism may be employed by PepT$_{Sh}$ to facilitate transport of modified peptides by replacing direct interactions between the peptide and the transporter when these are sterically restricted, as we observe with *S*-Cys-Gly-3M3SH. The interaction of the amino termini of peptides with both the mammalian and bacterial POT family transporters has been identified as facilitating high affinity binding (*Weitz et al., 2007*; *Meredith et al., 2000*). The reduced IC$_{50}$ value for the modified peptide in PepT$_{Sh}$ (*Figure 3B*), may therefore be a combination of steric restraints resulting in the accommodation of the extended side chain group, and the subsequent knock on effects on the position of the interactions between the amino and carboxy groups with conserved side chains in the binding site.

## Functional characterisation of PepT$_{Sh}$ binding site

S-Cys-Gly-3M3SH is the first human derived peptide with a modified side chain to be captured in a POT family transporter, and contains biochemical features similar to physiological peptides (Cys-Gly), but also incorporates the non-peptide thioalcohol group (3M3SH). We therefore wished to explore whether S-Cys-Gly-3M3SH is recognised using a similar mechanism to physiological peptides. Several mutations in the binding site that have previously been shown to play an important role in peptide recognition within the bacterial POT family were therefore generated (*Newstead, 2017*). Tyrosine 41 forms part of a conserved E[33]xxERFxYY[41] motif on TM1 (*Solcan et al., 2012*), which plays a role in proton coupling and influences peptide uptake in several POT family members (*Doki et al., 2013*; *Ernst et al., 2009*; *Guettou et al., 2014*; *Lyons et al., 2014*). In PepT$_{Sh}$ Tyr41 hydrogen bonds with the carboxyl terminus of the terminal glycine amino acid in S-Cys-Gly-3M3SH (*Figure 4C,D*). Mutating Tyr41 to phenylalanine or alanine severely reduced transport of di-alanine compared to WT protein (*Figure 5A*). A similar negative effect on transport was observed in the related POT family transporter PepT$_{St}$, from *Streptococcus thermophilus* (*Solcan et al., 2012*), suggesting a common role for the conserved tyrosine in peptide recognition within the POT family. We then used the same variants in our whole cell assay for 3M3SH production. Interestingly, while both Tyr41Phe and Tyr41Ala variants had reduced activity, at about 50% of the wild-type protein (*Figure 5B*), the phenotype was not as severe as that observed in the liposome based assay. We

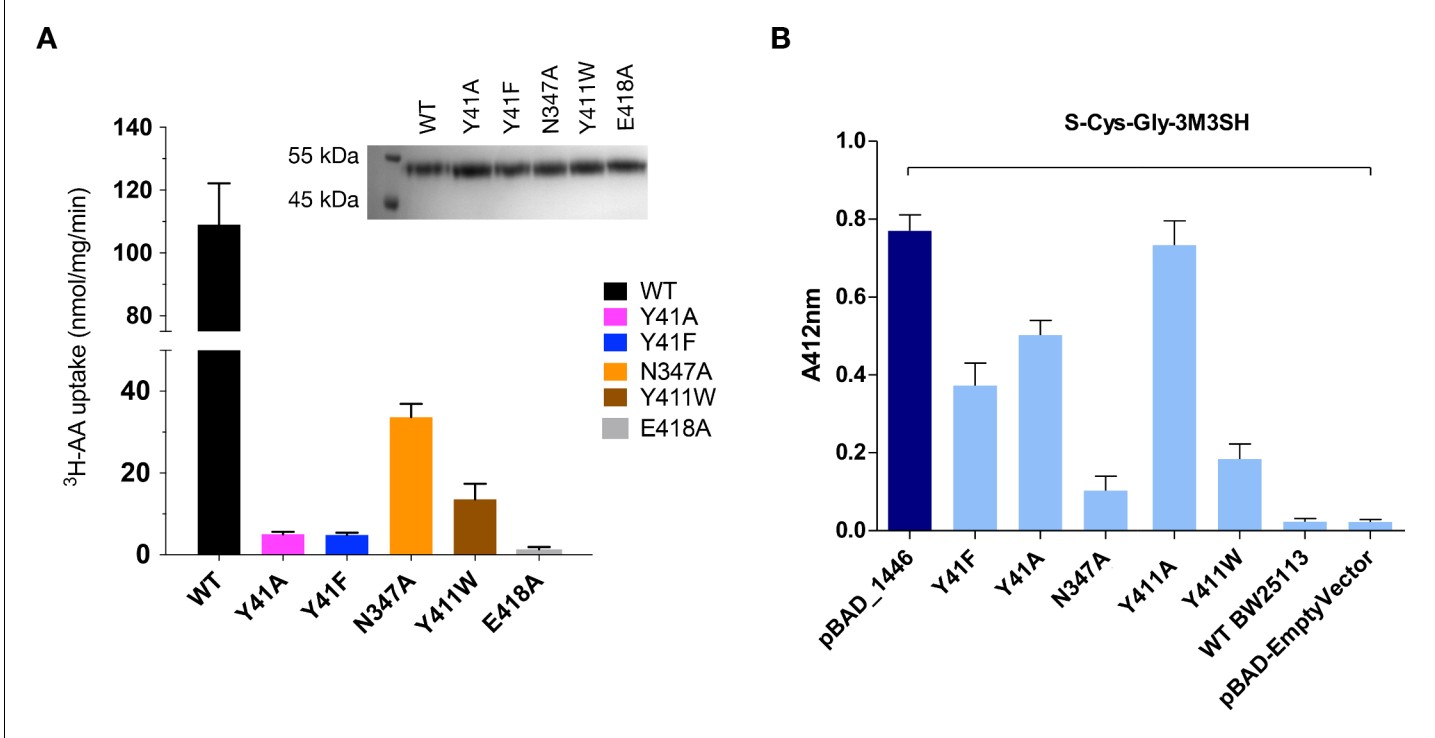

**Figure 5.** Functional characterisation of binding site residues. (**A**) Bar chart showing the di-alanine uptake for the WT and mutant variants of PepT$_{Sh}$. Inset – SDS-PAGE analysis of the reconstituted proteins used in the assay. (**B**) Biotransformation of *S*-Cys-Gly-3M3SH by *E. coli* K-12 (BW25113) overexpressing WT PepT$_{St}$ (pBAD_1446), or variants indicated, compared to background uptake in non-transformed cells (WT BW25113) and cells harbouring empty plasmid (pBAD-EmptyVector). Error bars: ±SD (biological triplicates).

DOI: https://doi.org/10.7554/eLife.34995.017

interpret this result to indicate that recognition of *S*-Cys-Gly-3M3SH is less dependent on this residue than di-Ala.

We next wanted to test the role of other residues we observed sitting close to the *S*-Cys-Gly-3M3SH ligand. In particular, we were interested in the contribution made by the Asn347 and Glu418, which coordinate a water molecule (W107) linking both the carboxyl and carbonyl groups of the ligand to the binding site (*Figure 4B*). The equivalent residues in homologous POT family transporters have been implicated in both peptide recognition (*Lyons et al., 2014*) and gating (*Solcan et al., 2012*), and in the case of Glu418 may facilitate the coupling between peptide recognition, proton movement and transport (*Parker et al., 2017*). Although a Glu418Ala variant had no activity in our assays, consistent with its proposed role in coordinating the intracellular gate (*Fowler et al., 2015*), we observed reduced uptake of di-Ala in the Asn347Ala variant (*Figure 5A*). Interestingly, this reduction was far more severe in the in vivo assay (*Figure 5B*), indicating that this interaction is more important for transport of the *S*-Cys-Gly-3M3SH peptide than di-Ala. This may reflect the requirement of Asn347 to coordinate the water molecule linking the peptide to the intracellular gate side chain Glu418, which in the case of the di-peptide may interact directly as in PepT$_{St}$, lessening the requirement for the Asn347 interaction.

The co-crystal structure also identified the acyl chain portion of the thioalcohol group was accommodated within an extended pocket, formed near the extracellular side of the binding site (*Figure 4E*). To test the importance of this pocket on *S*-Cys-Gly-3M3SH transport, we made a Tyr411Trp variant, which would block the cavity. Uptake was noticeably reduced, but still measurable at ~19% compared to WT (*Figure 5A*). The same mutation resulted in a similar large decrease in the biotransformation of *S*-Cys-Gly-3M3SH, while a Tyr411Ala mutation had wild-type biotransformation rates (*Figure 5B*). It is clear that the hydrophobic pocket in PepT$_{Sh}$ plays an important role in facilitating the recognition of the thioalcohol group in the binding pocket and, as discussed below, is a potential site for inhibitor design.

Taken together the current data lend further support to our model of an adaptable binding site within the POT family that can accommodate chemically diverse peptides and their derivatives through the use of both side chain and water mediated interactions with conserved residues and different specificity pockets to accommodate the different chemical side chains.

## Discussion

The production of malodour by humans is an evolutionary ancient olfactory process that likely was involved in mate selection and evolutionary processes that relied on chemical signalling (*Stoddart, 1990*). Evidence for the microbial origins of chemical signalling molecules is also seen in other animals such as foxes, deer, cats and beetles (*Wyatt, 2003*), but the details of this process in humans is less well understood. The thioalcohol 3M3SH, a major component of body odour, is produced in the human axilla as a glutathione-conjugated precursor and is processed to a unique dipeptide structure (*Figure 1*). The discovery that axilla-isolated strains of *S. hominis*, *S. lugdunensis* and *S. heamolyticus* are able to produce 3M3SH prompted the search for the molecular basis for this process (*Bawdon et al., 2015*). While S-Cys-Gly-3M3SH is a highly modified and uniquely human dipeptide, we reasoned that this moiety might dictate the uptake mechanism into malodour producing bacteria and here we identify a peptide transporter, PepT$_{Sh}$ (SH1446), from the POT family of secondary active transporters as being responsible for this process. Each of these strains have a POT family transporter that is closely related to PepT$_{Sh}$ (*Figure 4—figure supplement 2*), and we would predict they will also be able to uptake S-Cys-Gly-3M3SH.

The genetic intractability of *S. hominis* dictated that we use a biochemical approach to demonstrate that SH1446 is likely the sole route of uptake of S-Cys-Gly-3M3SH. First, this staphylococci has a similar repertoire of peptide transporters in *S. aureus*, in that it has a single POT transporter and one Opp ABC transporter, and previous work in *S. aureus* has demonstrated that the POT transporter, DtpT, is essential for growth on dipeptides and some tripeptides, while the ABC transporter (Opp3) has a function in transporting longer oligopeptides (*Hiron et al., 2007*). The use of L-Ala containing peptides as inhibitors in this study demonstrates an inhibitory effect on 3M3SH production that is consistent with the properties of a POT family transporter. Several attempts were made to genetically disrupt SH1446 in *S. hominis*, using tools developed for *S. aureus* and *S. epidermidis*, but these were unsuccessful, most likely due to the presence of multiple species-specific restriction modification (RM) systems which render this organism, as with many other coagulase negative (CoNS) species, genetically intractable.

The specific role for POT transporters in the transport of these unusually modified dipeptides is supported from the inability of the *E. coli* ABC transporters for oligopeptides and dipeptides, Opp and Dpp, to recognise and transport S-Cys-Gly-3M3SH. Given that S-Cys-Gly-3M3SH is effectively a dipeptide with a highly extended R-group on the first amino acid, it is noteworthy that only the promiscuous nature of the POT binding site is able to recognise this ligand. For OppA and DppA, the substrate binding proteins that initially recognise peptides transported by the ABC systems, the binding pocket is well defined and makes general main chain-main chain interactions to ensure that the protein can bind peptides in a sequence independent manner (*Tame et al., 1995*; *Sleigh et al., 1997*; *Sleigh et al., 1999*). However, the pocket has evolved to accommodate regular amino acid side chains and the unusually large structure of the thioalcohol attachment would likely be sterically incompatible with recognition by either OppA or DppA. The data presented here further support the idea that POT transporters have an unusually promiscuous binding site that may play an evolutionary advantageous role in accepting modified peptides as substrates.

Recently, the human POT family transporters, PepT1 and PepT2, have gained additional attention due to their role in the absorption and retention of several important classes of drug molecule (*Smith et al., 2013*), including peptide modified pro-drugs. Similar to S-Cys-Gly-3M3SH, these pro-drugs are drug molecules conjugated to a peptide or amino acid (*Brandsch, 2013*). Pro-drugs are able to compete with natural peptides for transport via PepT1 and PepT2, and are therefore actively transported into the body (*Ganapathy et al., 1998*; *Gupta et al., 2013*; *Terada and Inui, 2012*). The promiscuous nature of the POT family binding site has been shown to derive, in part, from their ability to recognise peptides in different conformations and through the availability of different specificity pockets, which are able to accommodate chemically different classes of peptides (*Newstead, 2017*). Our results here provide the first evidence that specificity pockets can also

accommodate exotic chemical extensions. The observation that water plays a role in facilitating the interaction between the S-Cys-Gly-3M3SH peptide and PepT$_{Sh}$ may also indicate a broader role for water in facilitating the recognition of xenobiotic peptides. In particular, the observation that water may bridge interactions between the binding site and either the amino or carboxy group of the ligand would greatly increase the number of conformations a peptide could adopt in the transporter and still trigger proton coupled transport.

Finally, we can ask the questions, why do these bacteria transport S-Cys-Gly-3M3SH and can this process be inhibited to provide effective malodour control? The presumed biochemical pathway for breakdown of the S-Cys-Gly-3M3SH is outlined in *Figure 6*. The firststep is the likely removal of the glycine by a dipeptidase and then the lyase reaction that cleaves the carbon-sulphur bond to release pyruvate and ammonia in addition to 3M3SH. The route of export of 3M3SH is unknown, but currently assumed to be diffusion across the bacterial inner membrane. The bacterium will gain a

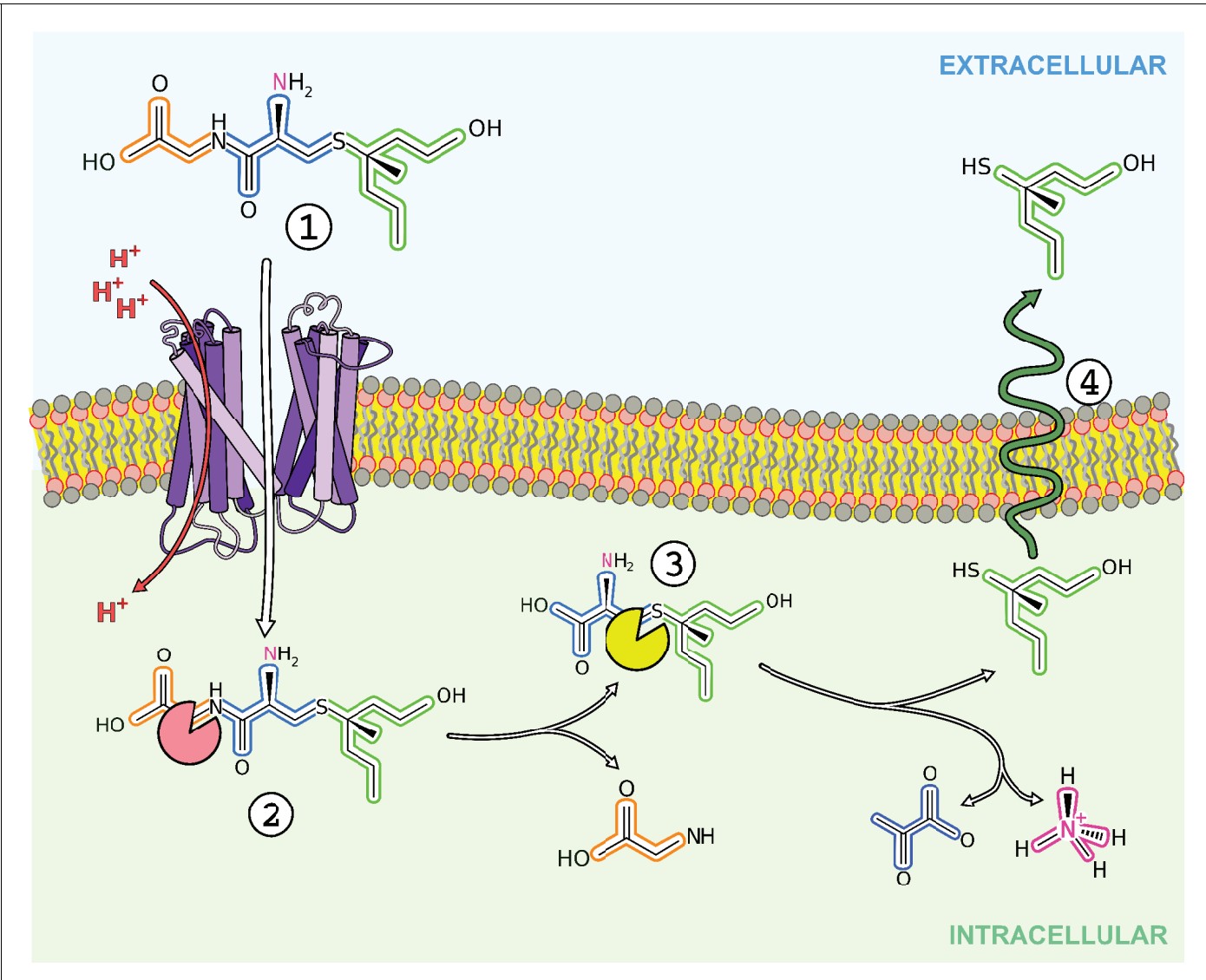

**Figure 6.** Overview of transport and intracellular biotransformation of *S*-Cys-Gly-3M3SH in *Staphylococcus hominis*. 1. Cys-Gly-(*S*)−3M3SH is actively transported across the membrane by the di/tri peptide transporter PepT$_{Sh}$ along with the movement of protons. 2. A dipeptidase cleaves the terminal glycine residue of *S*-Cys-Gly-3M3SH producing Cys-(*S*)−3M3SH. 3. The thiol is liberated from Cys-(S)−3M3SH by a C-S β-lyase that also releases ammonia and pyruvate, the volatile thiol (3M3SH) then diffuses or is exported out the cell. 4.
DOI: https://doi.org/10.7554/eLife.34995.018

nutritional benefit from the catabolism of S-Cys-Gly-3M3SH via the release of both nitrogen and carbon as glycine, pyruvate and ammonia (*Figure 6*), which may have been the evolutionary trigger for the adaptation to catabolise this chemical. Asian individuals with single nucleotide polymorphisms (SNPs) in ABCC11, the transporters involved in the secretion of S-Cys-Gly-3M3SH (*Figure 1*) have been studied (*Nakano et al., 2009*; *Baumann et al., 2014b*). Recently, a microbiological investigation of individuals with wild-type alleles of *ABCC11* compared to heterozygous and double mutant genotypes revealed that the proportion of *Staphylococcus* sp. is higher in the underarms of humans that produce the precursor (*Harker et al., 2014*). While these studies are limited in that they do not provide species level identification, they are at least consistent with the idea that the production of the precursor positively correlates with the presence of *Staphylococcus* sp. in the underarm. The discovery of the transporter and molecular understanding of the mode of S-Cys-Gly-3M3SH binding now suggests a specific route to develop inhibitors of malodour production by directly inhibiting the transporter SH1446 (*Grice, 2014*). Importantly, such intervention would not impact on the overall axillar microbiome, but instead specifically target the reduction in body odour production.

# Materials and methods

**Key resources table**

| Reagent type (species) or resource | Designation | Source or reference | Identifiers | Additional information |
|---|---|---|---|---|
| Ggene (*Escherichia coli*) | *dtpA* | BW25113 | b1634, ECK1630 | POT transporter family |
| Gene (*Escherichia coli*) | *dtpB* | BW25113 | b3496, ECK3481 | POT transporter family |
| Gene (*Escherichia coli*) | *dtpC* | BW25113 | b4130, ECK4124 | POT transporter family |
| Gene (*Escherichia coli*) | *dtpD* | BW25113 | b0709, ECK0698 | POT transporter family |
| Gene (*Escherichia coli*) | *tnaA* | BW25113 | b3708, ECK3701 | Tryptophanase |
| Gene (*Staphylococcus hominis*) | *SH1446* | SK119 | STAHO0001_1446 (GenBank: EEK12089.1) | POT transporter family |
| Gene (*Staphylococcus hominis*) | *SH0415* | SK119 | STAHO0001_0415 (GenBank: EEK11635.1) | MFS transporter family |
| Strain, strain background (*Escherichia coli*) | BW25113 | *Baba et al., 2006* | | F-, Δ(*araD-araB*)567, Δ*lacZ*4787(::*rrnB*-3), lambda-, rph-1, Δ(*rhaD-rhaB*)568, *hsdR*514 |
| Strain, strain background (*Escherichia coli*) | JW1237 | *Baba et al., 2006* | | Δ*oppC::kanR* |
| Strain, strain background (*Escherichia coli*) | JW3511 | *Baba et al., 2006* | | Δ*dppC::kanR* |
| Strain, strain background (*Escherichia coli*) | JW1626 | *Baba et al., 2006* | | Δ*dtpA::kanR* |
| Strain, strain background (*Escherichia coli*) | JW3463 | *Baba et al., 2006* | | Δ*dtpB::kanR* |
| Strain, strain background (*Escherichia coli*) | JW4091 | *Baba et al., 2006* | | Δ*dtpC::kanR* |

*Continued on next page*

Continued

| Reagent type (species) or resource | Designation | Source or reference | Identifiers | Additional information |
|---|---|---|---|---|
| Strain, strain background (Escherichia coli) | JW0699 | *Baba et al., 2006* | | Δ*dtpD::kanR* |
| Strain, strain background (Escherichia coli) | JW3686 | *Baba et al., 2006* | | Δ*tnaA::kanR* |
| Strain, strain background (Escherichia coli) | JW2975 | *Baba et al., 2006* | | Δ*metC::kanR* |
| Strain, strain background (Escherichia coli) | JW1614 | *Baba et al., 2006* | | Δ*malY::kanR* |
| Strain, strain background (Escherichia coli) | JW1285 | *Baba et al., 2006* | | Δ*sapC::kanR* |
| Strain, strain background (Escherichia coli) | JW0590 | *Baba et al., 2006* | | Δ*cstA::kanR* |
| Strain, strain background (Escherichia coli) | ΔoppC | This study. KanR removed from parent strain | | Δ*oppC* |
| Strain, strain background (Escherichia coli) | ΔdppC | This study. KanR removed from parent strain | | Δ*dppC* |
| Strain, strain background (Escherichia coli) | ΔdtpA | This study. KanR removed from parent strain | | Δ*dtpA* |
| Strain, strain background (Escherichia coli) | ΔdtpB | This study. KanR removed from parent strain | | Δ*dtpB* |
| Strain, strain background (Escherichia coli) | ΔdtpC | This study. KanR removed from parent strain | | Δ*dtpC* |
| Strain, strain background (Escherichia coli) | ΔdtpD | This study. KanR removed from parent strain | | Δ*dtpD* |
| Strain, strain background (Escherichia coli) | ΔtnaA | This study. KanR removed from parent strain | | Δ*tnaA* |
| Strain, strain background (Escherichia coli) | ΔDB1 | This study. KanR removed from parent strain | | Δ*oppC* Δ*dppC* |
| Strain, strain background (Staphylococcus hominis) | SK119 | | Taxonomy ID: 629742 (Accession: PRJNA34087) | |
| Strain, strain background | C43(DE3) | *Miroux and Walker, 1996* | | |
| Strain, strain background | pWaldo-GFPe | *Drew et al., 2001* | | |
| Genetic reagent (Escherichia coli) | pBADcLIC2005 | *Geertsma and Poolman, 2007* | | pBAD vector modified with LIC cassette, P$_{BAD}$ |
| Genetic reagent (Escherichia coli) | pKD46 | *Datsenko and Wanner, 2000* | | Contains Kan$^R$ cassette |

*Continued on next page*

*Continued*

| Reagent type (species) or resource | Designation | Source or reference | Identifiers | Additional information |
|---|---|---|---|---|
| Genetic reagent (Escherichia coli) | pCP20 | *Cherepanov and Wackernagel, 1995* | | FLP$^+$, pSC101 origin, λ cl857+, λ p$_R$ Rep$^{ts}$, Amp$^R$, Cm$^R$ |
| Recombinant DNA reagent | pBADcLIC-dtpA | This study. Cloned using primers in *Supplementary file 1* | | |
| Recombinant DNA reagent | pBADcLIC-dtpB | This study. Cloned using primers in *Supplementary file 1* | | |
| Recombinant DNA reagent | pBADcLIC-dtpC | This study. Cloned using primers in *Supplementary file 1* | | |
| Recombinant DNA reagent | pBADcLIC-dtpD | This study. Cloned using primers in *Supplementary file 1* | | |
| Recombinant DNA reagent | pBADcLIC-SH0415 | This study. Cloned using primers in *Supplementary file 1* | | |
| Recombinant DNA reagent | pBADcLIC-SH1446 | This study. Cloned using primers in *Supplementary file 1* | | |
| Peptide, recombinant protein | SH1446 | This study. Purified from *E. coli* BW25113 containing pBADcLIC-SH1446 | | |
| Peptide, recombinant protein | TnaA | This study. Purified from *E. coli* BW25113 containing pBADcLIC-TnaA | | |
| Chemical compound, drug | S-Cys-Gly-3M3SH | Peakdale Molecular (UK) | | Custom made malodour substrates |
| Chemical compound, drug | L-Ala | Sigma-Aldrich (St. Louis, MO, USA) | A7627 | |
| Chemical compound, drug | L-Ala-Ala | Sigma-Aldrich (St. Louis, MO, USA) | A9502 | |
| Chemical compound, drug | L-Ala-Ala-Ala | Sigma-Aldrich (St. Louis, MO, USA) | A9627 | |
| Chemical compound, drug | L-Ala-Ala-Ala-Ala | Cambridge Biosciences | H-1260.0250 | |
| Chemical compound, drug | n-Decyl-β-D-maltopyranoside | Anatrace | D322LA | |
| Chemical compound, drug | n-Dodecyl-β-D-maltoside | Anatrace | D310LA | |
| Chemical compound, drug | Nickel-NTA Superflow resin | ThermoFisher | 88221 | |

## Bacterial strains, media and plasmids

*Escherichia coli* K-12 was used as the model organism throughout this study. *Escherichia coli* K-12 single gene deletions were obtained from the in-house *E. coli* Keio collection at University of York. Bacterial strains and plasmids used in this study are listed in *Supplementary file 1* and *2*. All strains were cultured in LB (1% tryptone, 0.5% yeast extract, 1% NaCl) unless otherwise stated, Antibiotics were used at the following concentrations Ampicillin 100 μg/mL and Kanamycin 50 μg/mL; L-arabinose concentrations are indicated. PCR, plasmid extractions, and transformations were carried out as per standard protocols. For routine cloning, genomic DNA was extracted from BW25113 and high fidelity KOD polymerase was used (Novagen). Correct clones were confirmed by DNA sequencing. Specific details of plasmid constructions are detailed below.

## Construction of multiple mutants in single keio deletions

Multiple gene knockouts were created in single keio deletion strains. To create the double KO strain DB1 ($\Delta$oppC $\Delta$dppC), the Kan$^R$ cassette was removed from a target gene using pCP20 (encoding Flp recombinase) at 30°C, loss of antibiotic cassette was confirmed by PCR and sequencing. The second gene to be disrupted was PCR amplified with 100 bp homology including the FRT flanked Kan$^R$ cassette from the single gene deletion strain in the keio collection. Targeted gene disruptions were created using λ Red recombination as previously described (*Datsenko and Wanner, 2000*).

## Plasmid constructions and protein purification: pBAD-dtpA, dtpB, dtpC, dtpC, tnaA

Overexpression of the *E. coli* POT transporters were cloned into pBADcLIC2005 vector under the control of arabinose. Full-length genes were amplified from *E. coli* BW25113 genomic DNA using high fidelity KOD polymerase (Merck) using the corresponding primers listed in *Supplementary file 3*. In silico analysis of the available *Staphylococcus hominis* SK1119 genome identified STAH0001_1446 (SH1446) as a POT family member which orthologues present in all sequenced staphylococci and a MFS transporter STAH0001_0415 (SH0415) present in only *S. hominis* and *S. haemolyticus*. SH0415 and SH1446 genes were amplified from *S. hominis* genomic DNA using high fidelity KOD polymerase (Merck) using primers listed in *Supplementary file 3*. All PCR products were purified using Wizard SV Gel and PCR Clean-Up System (Promega), cloned into pBADcLIC2005 vector using standard ligation-independent cloning (LIC) techniques. All fragments were confirmed by PCR and DNA sequencing (GATC Biotech).

## Site directed mutagenesis of SH1446

The wildtype STAHO0001_1446 (SH1446) cloned onto the expression plasmid pBADcLIC2005 was used to individually generate the Y411W, Y411A, Y41A, Y41F, N167A and N347A mutants. To introduce the mutation, high fidelity inverse PCR was performed using divergent primers with one per pair being mutagenic (*Supplementary file 3*). The resultant products were then circularised by performing a blunt end ligation and transformed into *E. coli* BW25113. The mutant sequences were verified by DNA sequencing (GATC Biotech).

## Malodour precursor biotransformation assay malodour precursor substrates

S-Benzyl-L-cysteinylglycine and (S)-(1-(2-hydroxyethyl)−1-methylbutyl)-L-cysteinylglycine (S-Cys-Gly-3M3SH) (Peakdale Molecular) were prepared at a stock concentration of 10 mM in M9 salts (6 g $Na_2HPO_4$ g/L, 3 g $KH_2PO_4$ g/L, 0.5 g NaCl g/L) and stored at 4°C.

## S-Benzyl-L-cysteinylglycine and S-Cys-Gly-3M3SH biotransformation assays using resting cells

*E. coli* were grown overnight in LB, cells were harvested by centrifugation at 3, 500 rpm 10 min and resuspended in sterile M9 salts. Where appropriate, *E. coli* strains harbouring inducible genes on the pBADcLIC2005 plasmid were pre-induced during overnight growth with 0.0001% L-arabinose. To a sterile Eppendorf the following was added; 2 mM substrate (*S*-Benzyl-L-cysteinylglycine or *S*-Cys-Gly-3M3SH), cells to an $OD_{650nm}$ of 5 and M9 salts up to 200 µL. Cells only (no added substrate) and substrate only (no added cells) reactions were also included in every individual experiment. The reactions were incubated at 37°C for up to 48 hr with samples taken at appropriate time points. Liberated thiols were quantified using either MTS/PMS or DTNB labelling as detailed below.

## (3-(4,5-dimethylthiazol-2-yl)−5-(3-carboxymethoxyphenyl)−2-(4-sulfophenyl)−2H-tetrazolium) (MTS) and phenazine methosulfate (PMS)-mediated chemical labelling of benzyl-mercaptan thioalcohol

Liberated benzyl mercaptan was labelled as follows, 500 µl from each reaction was taken at the appropriate time point and centrifuged at 13 000 r.p.m. for 2 min. To chemically label benzyl-mercaptan, 50 µl of labelling reagent, consisting of 4.09 mM 3-(4,5-dimethylthiazol-2-yl)−5- (3-carboxymethoxyphenyl)−2-(4-sulfophenyl)−2H-tetrazolium (MTS) (Promega) and 0.2 mM phenazine methosulfate (PMS) (Sigma) (*Goodwin et al., 1995*), dissolved in dH$_2$O, was aliquoted into either a

96 well microplate or 0.5 ml Eppendorf tubes. For reactions using a 96 well microplate, 200 μl reaction supernatant was added to the pre- aliquoted 50 μl MTS/PMS. The plate was sealed with a semi-permeable film lid, incubated at room temperature for 50 min and the $A_{492nm}$ recorded on a Biotek PowerWave XS microplate spectrophotometer. The data was captured by KC junior and exported to Microsoft Excel. For reactions in Eppendorf tubes, 20 μl - 50 μl reaction supernatant was added to 50 μl MTS/PMS. The reaction was incubated at room temperature for 50 min. 20 μl was subsequently removed and diluted in 980 μl $dH_2O$ in a 1.5 ml disposable plastic cuvette. The $A_{492nm}$ was recorded.

## 5,5'-dithiobis-(2-nitrobenzoic acid) (DTNB)-mediated chemical labelling of 3M3SH

To quantify free 3M3SH, a 75 μl volume from each reaction (isolated as described above) was taken at the appropriate time point and centrifuged at 13, 000 rpm for 2 min. 950 μl labelling solution, consisting of: 800 μl $dH_2O$, 100 μl UltraPure Tris (pH 8.0) and 50 μl DTNB (*Ellman, 1958*) stock solution (50 mM sodium acetate, 2 mM DTNB, dissolved in $dH_2O$) was added to a 1.5 ml disposable cuvette. 50 μl reaction supernatant was added to the cuvette, mixed by pipetting and incubated at room temperature for 5 min. The $A_{412nm}$ was recorded on a Jenway 6305 spectrophotometer. (*Van Horn, 2003*). Where appropriate, E. coli strains harbouring inducible genes on the pBAD-cLIC2005 plasmid were pre-induced during overnight growth with 0.0001% L-arabinose. In all experiments cells only (no added substrate) and substrate only (no added cells) were also included.

## Protein expression and purification

The gene encoding *S. hominis* SH1446 was cloned into a C-terminal octa histidine GFP fusion vector (pWaldo-GFPe) (*Drew et al., 2006*) and subsequently transformed into *Escherichia coli* C43(DE3) cells (*Miroux and Walker, 1996*). Mutations were introduced using site directed mutagenesis followed by DpnI digestion. An overnight starter culture was used to inoculate 4 L of Terrific Broth (TB). Cells were left to grow at 37°C and aerated by shaking at 250 rpm. Overexpression of the fusion protein was induced when the culture reached an $OD_{600}$ of ~0.6, by adding isopropyl-β-D-thiogalactopyranoside (IPTG) to a final concentration of 0.4 mM. Following induction, the temperature was reduced to 25°C. Cells were harvested 16 hr later, and resuspended in phosphate buffered saline (PBS) containing deoxyribonuclease I from bovine pancreas, and stored at −80°C.Unless stated, the following steps were carried out at 4°C. Cells were thawed and disrupted at 32 kpsi (Constant Systems, UK). Cellular debris was removed by centrifugation at 30,000 x *g* for 30 min, and membranes were isolated by ultracentrifugation at 230,000 x *g* for 120 min. Membranes were resuspended in PBS using a dounce homogeniser, flash frozen, and stored at −80°C. Thawed membranes were diluted with 1x PBS (10 mL per gram of membrane), 20 mM imidazole (pH 8.0), 150 mM NaCl, and 1% n-Dodecyl-β-D-maltoside (DDM) and left to solubilise for 60 min, stirring. Non-solubilised membranes were removed by ultracentrifugation at 230,000 *g* for 60 min. Nickel-NTA resin (Thermo-Fisher Scientific) was added to the sample, using a ratio of 1 mL resin per gram of membrane, and left to bind, stirring, for approximately 90 min. The resin was then packed into a glass econo-column (BioRad, USA) under gravity. The resin was washed with 20 column volumes (CV) of wash buffer (1x PBS, 20 mM imidazole (pH 8.0), 150 mM NaCl and 0.05% DDM). The resin was further washed with 10CV of wash buffer containing 30 mM imidazole. The bound GFP-SH1446 fusion protein was eluted with wash buffer containing 250 mM imidazole. The eluent was combined with an equivalent concentration of hexa histidine tagged TEV protease and dialysed overnight against 20 mM Tris-HCl (pH 7.5), 150 mM NaCl, and 0.03% DDM. The cleaved protein was filtered using a 0.22 μm filter (Millipore, USA) and passed through a Ni-NTA 5 mL HisTrap column (GE Healthcare) to remove uncleaved GFP-SH1446 protein and TEV protease. The flow through was collected and concentrated to 300 μl using a 50 kDa MWCO concentrator (Vivapsin 20, Sartorius AG) and applied to a size exclusion column (Superdex 200 10/30, GE Healthcare) pre-equilibrated with 20 mM Tris-HCl (pH 7.5), 150 mM NaCl, and 0.03% DDM. Fractions containing purified SH1446 protein were pooled and concentrated to a final concentration of 15–20 mg.ml$^{-1}$.

For expression and purification of TnaA, overnight cultures were grown in LB at 37°C, 200 rpm. 1 L of LB was inoculated 1:50 with the starter culture and grown to an $OD_{600nm}$ ~0.6–0.8, target protein expression was induced with 0.001% arabinose. Following induction the culture was further

incubated at 37°C for ~8 hr. Cells were harvested by centrifugation at 4,500 rpm for 15 min and resuspended in 35 mL resuspension buffer (50 mM Kpi, 20% Glycerol, 200 mM NaCl and 10 mL Imidazole, pH 7.8) and stored at −80°C. All subsequent steps were carried out at 4°C, for protein purification cells were thawed and lysed using sonication with 3 s pulsed and 7 s pause for 3mins. Lysates were clarified by centrifugation 25,000 rpm ~30 min. The clarified supernatant was loaded onto a 5 mL HisTrap column (GE Healthcare) and affinity purified on an AKTA Start (GE Healthcare) using a standard protocol as per manufacturer's instructions.

## Protein crystallisation

Crystals of SH1446 were obtained *in-meso* phase. Briefly, concentrated protein was mixed with 9.9 monoacylglycerol (Monoolein, Sigma-Aldrich) at a protein:lipid ratio of 2:3, in a mechanical syringe mixer at 20°C (*Ai and Caffrey, 2000*). 50 nL of mesophase and 800 nL of precipitant solution was dispensed on to a 96-well glass plate using an *in meso* robot (Gryphon, Art Robbins Instruments). SH1446 S-Cys-Gly-3M3SH complex was crystallised in precipitant solution containing 26–27% (v/v) PEG 200, 220 mM $(NH4)_2HPO_4$, 110 mM sodium citrate (pH 5.0) and 5 mM S-Cys-Gly-3M3SH (Peakdale Molecular) and left overnight at 4°C, prior to crystallisation. Crystals appeared after 2–3 days, and grew to its maximum size after 10 days. Crystallisation wells were accessed using a tungsten-carbide glasscutter, and crystals were directly mounted onto 50–100 µm microloops (MiTeGen) and flash frozen at 100 K.

X-ray diffraction data were collected at ID23-2 at the European Synchrotron Radiation Facility on a Pilatus3 2M detector with an exposure setting of 0.05 s with a 0.2° oscillation range over 360° Diffraction data was indexed and integrated in XDS (*Kabsch, 2010*) and merged in Aimless (*Evans, 2011*).

## Structure solution and refinement

The ligand bound complex was solved by molecular replacement in PHASER (*McCoy et al., 2007*) using a modified search model of the bacterial peptide transporter from *Streptococcus thermophilus*, $PepT_{St}$ (*Lyons et al., 2014*), PDB: 4D2B. Iterative rounds of manual model building in COOT (*Emsley et al., 2010*) followed by refinement in autoBUSTER (Bricogne G. 2017) followed by refinement in Phenix (*Adams et al., 2010*), resulted in a final model, which was validated using the Molprobity server (*Chen et al., 2010*). Additional difference density was observed in the binding site and matched the stereo chemical properties and size of S-CG3M3SH. Ligand restraints were generated using the Grade Web Server (http://grade.globalphasing.org). Figures were prepared using PyMOL (Schrödinger, LLC).

## Reconstitution into proteoliposomes and transport assays

Purified SH1446 was detergent exchanged into n-Decyl-β-D-maltopyranoside (DM, Anatrace) using a size exclusion column (Superdex 200 10/30, GE Healthcare) pre-equilibrated with 20 mM Tris-HCl (pH 7.5), 150 mM NaCl, and 0.3% DM. Lipids (POPE:POPG in a 3:1 ratio) were mixed with the eluted protein at a ratio of 60:1. The protein:lipid mixture was rapidly diluted with 50 mM $KPO_4$ (pH 7.0) buffer to below the CMC value of DM. Proteoliposomes were harvested by ultracentrifugation (234,000 *xg*) for 120 min. The pellet was resuspended in 50 mM $KPO_4$ and dialysed overnight against 3L of 50 mM $KPO_4$ (pH 7.0). Proteoliposomes were recovered and subjected to three rounds of freeze thawing before storage at −80°C.

Proteoliposomes were thawed and harvested by ultracentrifugation at 150,000 *g*. The resulting pellet was carefully resuspended in INSIDE buffer (120 mM KCl, 2 mM $MgSO_4$, 5 mM HEPES pH 6.8) and 1 mM pyranine (trisodium 8-hydroxypyrene-1,3,6-trisulfonate). Proteoliposomes underwent eleven freeze-thaw cycles in liquid nitrogen before the sample was extruded through a 0.4 µm polycarbonate membrane and harvested by ultracentrifugation as before. Excess pyranine dye was removed with a Microspin G-25 column (GE Healthcare) pre-equilibrated with INSIDE buffer without pyranine. For the assay, proteoliposomes were diluted into OUTSIDE buffer (120 mM NaCl, 5 mM HEPES (pH 6.8), 2 mM $MgSO_4$).

A Cary Eclipse fluorescence spectrophotometer (Agilent Technologies) equipped with a cuvette and magnetic flea was used to carry out the transport assay. Dual fluorescence excitation was set to 460/415 nm with emission at 510 nm. Transport was initiated with 1 µM valinomycin, the resulting

data was exported and analysed using Prism (GraphPad Software). Transport data was normalised to one to facilitate comparison with data collected from multiple conditions.

## Competition assays

Proteoliposomes were harvested as before, the resulting pellet was resuspended in INSIDE buffer, subjected to three freeze-thaw cycles in liquid nitrogen, extruded through a 0.4 μm polycarbonate membrane and harvested by ultracentrifugation once more. For the assay, proteoliposomes were diluted in external buffer (110 mM NaCl, 10 mM $NaPO_4$ and 2 mM $MgSO_4$) supplemented with 50 μM di-alanine peptide containing trace amounts of tritiated $^3$H-di-alanine (specific activity 30 Ci/mmol) and the competing substrate at increasing concentrations as indicated. Competition assays were performed at 30°C, samples were taken at specified time points and transport was stopped by addition of 2 mL $H_2O$. Proteoliposomes were immediately filtered onto a 0.22 μm cellulose filter (Merck Millipore) using a vacuum manifold and subsequently washed twice more with 2 mL $H_2O$. The amount of peptide transported into the liposomes was calculated based on the specific activity for each peptide as detailed by the manufacturer and counting efficiency for the radioisotope in Ultima Gold (PerkinElmer) counted in a Wallac scintillation counter ($^3$H, 45% counting efficiency). Transport assays were performed a minimum of three times to generate an overall mean and S.E.M.

## Data availability

Atomic coordinates for the atomic models have been deposited in the Protein Data Bank under accession numbers 6EXS.

# Acknowledgements

We thank the staff of beamlines I24 Diamond Light Source, UK, and ID23eh2 at the European Synchrotron Radiation Facility (ESRF) for assistance. This work was supported by a Wellcome award (102890/Z/13/Z) to SN and BBSRC LINK funding to GHT (BB/N006615/1) and was initiated by CBMNnet proof of concept funding (BB/L013703/1) to GHT and SN. DB was supported by a BBSRC CASE studentship (BB/H016201/1).

# Additional information

## Competing interests

A Gordon James: affiliated with Unilever Discover. The author has no financial interests to declare. The other authors declare that no competing interests exist.

## Funding

| Funder | Grant reference number | Author |
|---|---|---|
| Wellcome | 102890/Z/13/Z | Simon Newstead |
| Biotechnology and Biological Sciences Research Council | BB/N006615/1 | Gavin H Thomas |
| Biotechnology and Biological Sciences Research Council | BB/L013703/1 | Gavin H Thomas Simon Newstead |
| Biotechnology and Biological Sciences Research Council | BB/H016201/1 | Daniel Bawdon A Gordon James Gavin H Thomas |

The funders had no role in study design, data collection and interpretation, or the decision to submit the work for publication.

## Author contributions

Gurdeep S Minhas, Reyme Herman, Michelle Rudden, Resources, Data curation, Formal analysis, Validation, Investigation, Visualization, Methodology, Writing—review and editing; Daniel Bawdon, Resources, Data curation, Formal analysis, Validation, Investigation, Visualization, Methodology;

Andrew P Stone, Investigation, Methodology; A Gordon James, Conceptualization, Resources, Formal analysis, Supervision, Funding acquisition, Investigation, Visualization, Methodology, Writing—original draft, Project administration, Writing—review and editing; Gavin H Thomas, Simon Newstead, Conceptualization, Resources, Data curation, Formal analysis, Supervision, Funding acquisition, Validation, Investigation, Visualization, Methodology, Writing—original draft, Project administration, Writing—review and editing

## Author ORCIDs

Gurdeep S Minhas (iD) http://orcid.org/0000-0003-4320-1243
Reyme Herman (iD) http://orcid.org/0000-0002-6620-3981
Michelle Rudden (iD) http://orcid.org/0000-0001-9617-087X
Andrew P Stone (iD) http://orcid.org/0000-0002-1087-9923
Gavin H Thomas (iD) http://orcid.org/0000-0002-9763-1313
Simon Newstead (iD) http://orcid.org/0000-0001-7432-2270

## Decision letter and Author response

Decision letter https://doi.org/10.7554/eLife.34995.027
Author response https://doi.org/10.7554/eLife.34995.028

# Additional files

## Supplementary files

• Supplementary file 1. List of strains used in this study
DOI: https://doi.org/10.7554/eLife.34995.019

• Supplementary file 2. List of plasmids used in this study
DOI: https://doi.org/10.7554/eLife.34995.020

• Supplementary file 3. List of oligonucleotides used in this study
DOI: https://doi.org/10.7554/eLife.34995.021

• Transparent reporting form
DOI: https://doi.org/10.7554/eLife.34995.022

## Data availability

Diffraction data have been deposited in the PDB under the accession code 6EXS. Vectors have been deposited in Addgene (deposit number 75759).

The following dataset was generated:

| Author(s) | Year | Dataset title | Dataset URL | Database, license, and accessibility information |
|---|---|---|---|---|
| Gurdeep SM, Newstead S | 2018 | Crystal structure of SH1446 in complex with S-Cys-Gly-3M3SH | www.rcsb.org/structure/6EXS | Publicly available at the RCSB Protein Data Bank (accession no: 6EXS) |

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
