## [Decision Letter]

Thank you for submitting your article "Structural basis of malodour precursor transport in the human axilla" for consideration by *eLife*. Your article has been reviewed by three peer reviewers, one of whom is a member of our Board of Reviewing Editors, and the evaluation has been overseen by Richard Aldrich as the Senior Editor. The reviewers have opted to remain anonymous.

The reviewers have discussed the reviews with one another and the Reviewing Editor has drafted this decision to help you prepare a revised submission.

The manuscript reports the identification and crystal structure determination of a transporter protein for S-Cys-Gly-3M3SH in *Staphylococcus hominis*. The protein is proposed to import the compound into the bacteria for the processing to 3M3SH, which is released as an underarm odor in humans. The crystallographic work is accompanied by transport and functional assays in whole cell *E. coli* and proteoliposomes. Flexibility of amino acid side chains at the substrate binding site is proposed to be the main factor that allows the transporter to recognize multiple types of peptide-derivatives. To identify the transporter, the authors side-step the genetic intractability of *Staphylococcus* spp. by transferring the relevant biochemical process to *E. coli* and then utilizing both the well characterized functional annotation and the knock-out libraries to dissect the pathways, moving back and forth between *S. hominis* and *E. coli* proteins.

The reviewers agreed that the work is important and should be seen by the broad readership of *eLife*. However, all reviewers agreed that it is not clear from the experiments presented whether the identified transporter is the only or even the main transporter of S-Cys-Gly-3M3SH in *S. hominis*. They felt that to make this conclusion, the authors would need to conduct a knockout experiment. Corresponding genetic tools do exist (e.g. PMID:27531908; PMID: 25104751; PMCID: PMC3709823).

Additional specific comments are:

1) The affinity for the modified peptide appears to be relatively low in high μM range. Is this affinity compatible with levels of peptide likely encountered under physiological conditions?

2) How evolutionarily broad is the ability of Staph. species to transport S-Cys-Gly-3M3SH? Are there PepT_Sh_ homologues that are likely unable to transport the peptide because of a mutated binding site? If yes, are these species still able to take up the peptide

3) Authors may want to be more transparent in the limitations of this study as the major components of this analysis are inferred from non-skin colonizing organisms. For example, the genomic searches were targeted and could not obviate additional proteins playing alternative pathways.

4) Does the 50% reduction of model substrate 5BLCG in *E. coli*-ΔdtpB cells suggest the existence of one more importer? Such an importer would probably help to explain the discrepancies seen with some mutant activity between proteoliposomes and whole cells. Does such an importer have a homolog in *Staphylococcus hominis*?

5) The average B-factor of the structure is 57.0, which is fairly low and indicates rigidity for the protein. If the residues at the substrate binding site are flexible as suggested by the authors, do they display higher B-factor values?

6) From the data presented, is it possible to estimate the turnover rate of the transporter?

[Editors' note: further revisions were requested prior to acceptance, as described below.]

Thank you for resubmitting your work entitled "Structural basis of malodour precursor transport in the human axilla" for further consideration at *eLife*. Your revised article has been favorably evaluated by Richard Aldrich (Senior Editor) and a Reviewing Editor.

There is only one small remaining issue. One of the reviewers has requested that you estimate the turnover rate of the transporter. You have provided the values in your response in nmol/min/mg. I believe that the reviewer wanted to see the numbers in mol/mol/min. It seems to me that this number comes up to approximately 1 mol/mol/min for the wild type transporter (assuming MW ca 50 kDa) unless I made an arithmetic error. This seems a bit slow. Could you please verify the turnover rate and include a comment on it in the manuscript.

---

## [Author Response]

The manuscript reports the identification and crystal structure determination of a transporter protein for S-Cys-Gly-3M3SH in Staphylococcus hominis. The protein is proposed to import the compound into the bacteria for the processing to 3M3SH, which is released as an underarm odor in humans. The crystallographic work is accompanied by transport and functional assays in whole cell E. coli and proteoliposomes. Flexibility of amino acid side chains at the substrate binding site is proposed to be the main factor that allows the transporter to recognize multiple types of peptide-derivatives. To identify the transporter, the authors side-step the genetic intractability of Staphylococcus spp. by transferring the relevant biochemical process to E. coli and then utilizing both the well characterized functional annotation and the knock-out libraries to dissect the pathways, moving back and forth between S. hominis and E. coli proteins.The reviewers agreed that the work is important and should be seen by the broad readership of eLife. However, all reviewers agreed that it is not clear from the experiments presented whether the identified transporter is the only or even the main transporter of S-Cys-Gly-3M3SH in S. hominis. They felt that to make this conclusion, the authors would need to conduct a knockout experiment. Corresponding genetic tools do exist (e.g. PMID:27531908; PMID: 25104751; PMCID: PMC3709823).

We thank for the reviewers for including these references, which are all genetic tools to manipulate *S. aureus*. This important pathogen has been studied for many decades and its genetic tractability has been facilitated by the generation of restriction/modification (RM) deficient strains such as RN4220 that will accept DNA from a wide range of sources. Even so, such DNA often requires passage through special *E. coli* strains modified to give the correct modification patterns that prevents their immediate degradation when transformed into *S. aureus*. Coagulase negative species of *Staphylococci* include the dominant skin bacterium *S. epidermidis*, for which a few of the tools developed for *S. aureus* can work, however *S. hominis* is poorly studied coagulase negative species of *Staphylococcus* for which no genetic tools have previously been tested or developed. We have tried extensively to genetically manipulate *S. hominis* for this study; however, due the presence of multiple RM systems (it contains at least 4 from our in silicoanalysis) the strain is extremely refractory to the introduction of any foreign DNA, an essential initial step for undertaking any genetic engineering. There is not a single example in the literature of a gene knock-out in this species and to develop the tools to do this is a major project in itself.

However, we fully appreciated the reviewers concern that the evidence that the identified transporter “is the only or even the main transporter of *S*-Cys-Gly-3M3SH in *S. hominis*” and we have undertaken significant additional experimentation to demonstrate this.

Our strategy is to examine the inhibitory profile of peptides to whole cell systems that are able to produce 3M3SH. In the initial submission, we did not clearly explain that *S. hominis* and other *Staphylcocci* only contain a single POT transporter (Figure 2—figure supplement 4). In addition to one POT they contain between 1-4 ABC transporters of the Opp family. An important experimental study on the relative function of these peptides transporters in *S. aureus* demonstrated that only one of the ABC transporters (Opp-3) was important for use of oligopeptides (from 3 to 8 amino acids), while the POT transporter was essential for growth on all dipeptides and some tripeptides. (Hiron et al., 2007). Therefore, we reasoned that if di- and tripeptides could inhibit the production of 3M3SH whereas free amino acids and longer peptides could not then this would be strong biochemical support that they were entering via the POT transporter that we characterised in this study.

Using whole-cell systems we measure the ability of different peptides to inhibit production of 3M3SH. In our *E. coli* system, where the expression of the *S. hominis* POT protein SH1446 is essential to observe 3M3SH production, the process is inhibited by both di- and tri-peptides but not L-Ala or tetra-Ala (presented as new data in Figure 2—figure supplement 4). The same four molecules were tested at the same concentrations against purified TnaA, the cytoplasmic enzyme that cleaves the peptide-linked thioalcohol to 3M3SH and they have no effect (Figure 2—figure supplement 5), demonstrating that the observed inhibition is at the level of transport.

We then repeated these experiments in *S. hominis* directly and observed a similar pattern. This is especially significant for the di-peptide, which specifically inhibits the POT transporter and not the ABC transporter, while we do also see inhibition with the tripeptide. The inhibition observed at the highest concentration of tetra-Ala is an artefact of the low pH required to keep the chemical in solution, which is impacting negatively on the *S. hominis* cells.

From these data we are now really confident that the SH1446 protein is the major route of CysGly-3M3SH uptake in *S. hominis*.

Additional specific comments are:1) The affinity for the modified peptide appears to be relatively low in high μM range. Is this affinity compatible with levels of peptide likely encountered under physiological conditions?

The physiological concentrations of S-Cys-Gly-3M3SH in the axilla are not known, so it is difficult to comment on this. We note that the enzyme that produces S-Cys-Gly-3M3SH from glutathionyl3M3SH, the human GGT1 enzyme, has been partially biochemically characterised (Baumann et al., 2014) and 1 mM substrate was used in their assays perhaps suggesting that the concentration of the precursors for secretion in the axilla cells is in this region, although a precise *K*_M_ for this enzyme was not determined. We would not expect the affinity of the transporter to be particularly high given that SH1446 is likely the main route by which the organism acquires dipeptides and some tripeptides, but speculate that it is high enough to be able to result in the small flux required to move S-Cys-Gly-3M3SH into the cell for its biotransformation.

2) How evolutionarily broad is the ability of Staph. species to transport S-Cys-Gly-3M3SH? Are there PepT_Sh_ homologues that are likely unable to transport the peptide because of a mutated binding site? If yes, are these species still able to take up the peptide?

The single POT transporter present in *S. hominis* is present in most *Staphylococi* and therefore we speculate that most other *Staphylococci* can transporter the precursor. In related work we have discovered a particular CS-lyase enzyme specific to the malodour producing species of *Staphylococci* that we speculate influences whether bacteria make 3M3SH or not. In our phylogenetic analysis of the POT transporters, we show that most *Staphylococci* species contain an orthologous *PepT_Sh_* system, with some CoNS stains possessing two proteins. All these proteins contain the conserved motif ExxERF/YY, essential for protonation and peptide binding (1). As outlined in the paper the conserved features of *PepT_Sh_* classify it as an archetypal POT family transporter, the unique hydrophobic pocket that specifically accommodates our physiologically relevant substrate allows for this di- tri peptide (DtpT) permease to transport modified dipeptides. Without detailed structural information, we are unable to compare this *PepT_Sh_* with closely related organisms.

3) Authors may want to be more transparent in the limitations of this study as the major components of this analysis are inferred from non-skin colonizing organisms. For example, the genomic searches were targeted and could not obviate additional proteins playing alternative pathways.

We recognise one of the major limitations of this study is the inability to genetically manipulate *S. hominis* to create functional knockouts. We have now provided more evidence in our *E. coli* model for the role of *PepT_Sh_*in S-Cys-Gly-3M3SH transport in terms of selectively inhibiting *PepT_Sh_*. The new data further supports the inhibition seen for both wild-type *S. hominis* and *E. coli* overexpressing *PepT_Sh_*. We are confident in our *in silico* analysis that *S. hominis* contains a single DtpT POT system, and that this is the peptide transport system required for the transport *S-*CysGly-3M3SH.

4) Does the 50% reduction of model substrate 5BLCG in E. coli-ΔdtpB cells suggest the existence of one more importer? Such an importer would probably help to explain the discrepancies seen with some mutant activity between proteoliposomes and whole cells. Does such an importer have a homolog in Staphylococcus hominis?

The 50% reduction, and not total loss of activity in the *E. coli-ΔdtpB* strain is not surprising given that *E. coli* contains 4 POT transporters, named DtpA-D, each with distinct and overlapping substrate specificities that are involved in di- and tri-peptide uptake (Figure 2B) (2) and so the residual activity is almost certainly through the activity of one of these systems. We have a separate publication in preparation on the physiological functions of these 4 transporters and can report that a strain with all 4 deleted has completely lost the ability to produce 3M3SH (Bawdon et al., in preparation). This data further supports our hypothesis that the POT transporters are the sole route of Cys-Gly-3M3SH entry into bacteria, remembering that *S. hominis* has only a single POT system (SH1446 in *S. hominis*) and there are not multiple homologs like in *E. coli*.

5) The average B-factor of the structure is 57.0, which is fairly low and indicates rigidity for the protein. If the residues at the substrate binding site are flexible as suggested by the authors, do they display higher B-factor values?

The flexibility we refer to in the manuscript concerned the ability of the binding site to recognise ligands in different binding modes and conformations, as demonstrated in the homologous transporter PepT_St_ (5). We did not intend to infer that the side chains are structurally flexible in the structure – or indeed, are any more flexible that in other related POT family transporters. Indeed, S-CG-3M3SH is well coordinated in the binding site and has similar B-factors to the surrounding side chains (80 vs 54 Å^2^). We have now revised the wording in the manuscript.

“Taken together the current data lend further support to our model of an adaptablebinding site within the POT family that can accommodate chemically diverse peptides and their derivatives through the use of both side chain and water mediated interactions with conserved residues and different specificity pockets to accommodate the different chemical side chains.**”**

6) From the data presented, is it possible to estimate the turnover rate of the transporter?

We have calculated the turnover number for the transporter as follows:

Wild-Type 189 ± 37 nmol/min/mg

Y41A: 7.2 ± 0.58

Y41F 6.5 ± 0.40

N347A 42.9 ± 0.30

Y411W 34.0 ± 3.45

All the variants we infer mechanistic information from are functional, although clearly some changes have more negative effects than others. We try, wherever practically possible to take into account changes in *K*_M_ values when undertaking assays on functionally affected variants, such as Y41A/F in the study.

References:

(1) Newstead S. 2017. Recent advances in understanding proton coupled peptide transport via the POT family. Curr Opin Struct Biol 45:17–24.

(2) Harder D, Stolz J, Casagrande F, Obrdlik P, Weitz D, Fotiadis D, Daniel H. 2008. DtpB (YhiP) and DtpA (TppB, YdgR) are prototypical proton-dependent peptide transporters of Escherichia coli. FEBS J 275:3290–3298.

(3) Yu D, Pi B, Yu M, Wang Y, Ruan Z, Feng Y, Yu Y. 2014. Diversity and evolution of oligopeptide permease systems in staphylococcal species. Genomics 104:8–13. (4) Hiron A, Borezée-Durant E, Piard J-C, Juillard V. 2007. Only One of Four Oligopeptide Transport Systems Mediates Nitrogen Nutrition in Staphylococcus aureus. J Bacteriol 189:5119–5129.

(5). Lyons J.A, Parker, J. et al. 2014. Structural basis for polyspecificity in the POT family of proton coupled oligopeptide transporters. EMBO Reports 15:886-893

[Editors' note: further revisions were requested prior to acceptance, as described below.]

The manuscript has been improved but there are some remaining issues that need to be addressed before acceptance, as outlined below:There is only one small remaining issue. One of the reviewers has requested that you estimate the turnover rate of the transporter. You have provided the values in your response in nmol/min/mg. I believe that the reviewer wanted to see the numbers in mol/mol/min. It seems to me that this number comes up to approximately 1 mol/mol/min for the wild type transporter (assuming MW ca 50 kDa) unless I made an arithmetic error. This seems a bit slow. Could you please verify the turnover rate and include a comment on it in the manuscript.

Thank you for giving us the opportunity to submit our revised manuscript titled “Structural basis of malodour precursor transport in the human axilla’. Our calculations put the turnover number at 10. The WT transport rate was determined as ~189 nmol/min/mg using 50uM di-alanine as substrate. The MW of PepT_Sh_ is 55361Da, giving 18.06 nmol in 1mg. So, 189/18 = 10.5. I have included this the paper after we describe the IC_50_ data:

“To understand the recognition of S-Cys-Gly-3M3SH compared to natural peptides, we determined IC50 values using competition with 3H-di-alanine. The IC_50_ value for tri-alanine, dialanine and Cys-Gly were 23.7 ± 3.4, 72.2 ± 9.7 and 115 ± 22.2 μM, respectively (Figure 3B), indicating that PepT_Sh_ is able to recognise tri-Ala more favourably than di-Ala. However, for S-Cys- Gly-3M3SH the IC_50_ value was 362 ± 21.8 μM, three-fold higher than the Cys-Gly peptide. The turnover number of PepT_Sh_ for di-peptide transport was calculated as ~10 per minute. Our data therefore show that while PepT_Sh_ can transport the modified peptide, the addition of the thioalcohol group on the cysteine side chain has substantially impaired the recognition and transport versus natural peptides.”

10 turnovers per minute is at the lower end of activity for the bacterial peptide transporters we have worked with. PepTSo in contrast is ~ 10x faster at 100 turnovers/minute for example. I am happy to comment on this if the referee thinks this may have an impact on the interpretation of our competition results. I must admit, we didn’t consider 10 to be abnormally slow when interpreting the assay results.